# DiffKGW: Stealthy and Robust Diffusion Model Watermarking

**Tianxin Wei**[1], **Ruizhong Qiu**[1], **Yifan Chen**[2], **Yunzhe Qi**[1], **Jiacheng Lin**[1], **Wenxuan Bao**[1], **Wenju Xu**[3], **Sreyashi Nag**[3], **Ruirui Li**[3], **Hanqing Lu**[3], **Zhengyang Wang**[3], **Chen Luo**[3], **Hui Liu**[3], **Suhang Wang**[4], **Jingrui He**[1], **Qi He**[5], **Xianfeng Tang**[3]

[1] **University of Illinois Urbana-Champaign**
[2] **Hong Kong Baptist University**
[3] **Amazon**
[4] **Pennsylvania State University**
[5] **Microsoft**

**Reviewed on OpenReview:** `https://openreview.net/forum?id=OXi9vcIOgD`

## Abstract

Diffusion models are known for their supreme capability to generate realistic images. However, ethical concerns, such as copyright protection and the generation of inappropriate content, pose significant challenges for the practical deployment of diffusion models. Recent work has proposed a flurry of watermarking techniques that inject artificial patterns into initial latent representations of diffusion models, offering a promising solution to these issues. However, enforcing a specific pattern on selected elements can disrupt the Gaussian distribution of the initial latent representation. Inspired by watermarks for large language models (LLMs), we generalize the LLM KGW watermark to image diffusion models and propose a stealthy probability adjustment approach DiffKGW that preserves the Gaussian distribution of initial latent representation. In addition, we dissect the design principles of state-of-the-art watermarking techniques and introduce a unified framework. We identify a set of dimensions that explain the manipulation enforced by watermarking methods, including the distribution of individual elements, the specification of watermark shapes within each channel, and the choice of channels for watermark embedding. Through the empirical studies on regular text-to-image applications and the first systematic attempt at watermarking image-to-image diffusion models, we thoroughly verify the effectiveness of our proposed watermark identification framework through comprehensive evaluations. On all the diffusion models, including Stable Diffusion, our approach induced from the proposed framework not only preserves image quality but also outperforms existing methods in robustness against a wide range of attacks.

## 1 Introduction

The rise of diffusion models has significantly impacted image generation, enabling the creation of diverse and high-quality images across various styles. However, the widespread use of these models also introduces critical ethical challenges, particularly concerning copyright protection and the generation of inappropriate or misleading content. In this regard, *watermarking* generated images offers a promising approach to tracing image origins and mitigating potential misuse.

A core challenge in image watermarking identification is the trade-off between the robustness of the watermark and the quality of the generated images. Traditional watermarking techniques (Al-Haj, 2007; Navas et al., 2008) primarily rely on post-processing methods to embed subtle modifications into the image's frequency

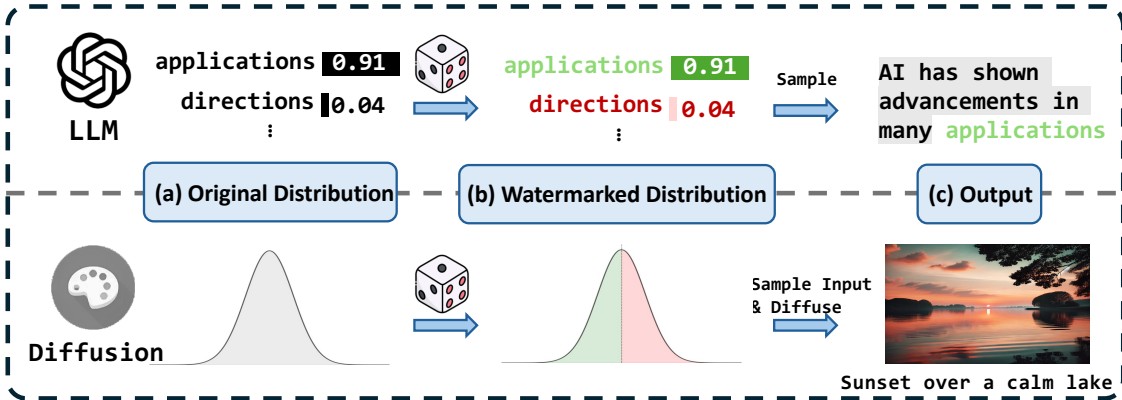

Figure 1: Illustration of the watermarking process, showing the relationship between LLMs and diffusion models. (a) Original Distribution: Initial probability distributions for both LLM and diffusion models. (b) Watermarked Distribution: Adjusted probabilities for selected elements, with a red/green split in the LLM echoed in the diffusion model's modified red/green sampling distribution. (c) Output: Sampled outputs with watermarks.

representation, making them imperceptible to human users. Though effective, these approaches suffer from reduced image quality and insufficient robustness against common attacks such as compression and cropping, to name a few.

To improve the trade-off, end-to-end deep learning methods (Zhang et al., 2019; Zhu et al., 2018; Hayes & Danezis, 2017) are proposed to construct watermarks with powerful learning capabilities, while such methods are black-box and require additional training. Specifically for diffusion models, latent-representation-based watermarking methods (Wen et al., 2023; Yang et al., 2024) have been developed, which embed watermarks by manipulating latent representations to match predefined patterns.

However, enforcing a specific pattern on selected elements can disrupt the independent and identically distributed (IID) Gaussian distribution of the initial latent representation. Inspired by watermarks for large language models (LLMs), we generalize the LLM KGW watermark (Kirchenbauer et al., 2023) to image diffusion models and propose a stealthy probability adjustment approach that preserves the Gaussian distribution of initial latent representation, as illustrated in Figure 1. In addition to that, the key components thereof are poorly understood, and the connections between watermarking methods remain unclear. In this paper, we carefully analyze state-of-the-art watermarking methods, and introduce a unified framework that identifies and connects the underlying design principles. This framework reveals three critical design dimensions: ❶ the distribution of individual elements, ❷ the specification of watermark shapes within each channel, and ❸ the choice of channels for embedding. Under the unified framework, we integrate key designs from both image and language watermarking; we propose a novel, training-free watermarking approach that applies directly to diffusion models without altering the training process. Notably, our method embeds watermarks directly into the latent space of diffusion models (rather than the frequency domain), avoiding extraneous operations and errors during detection. Extensive experiments are conducted; in addition to conventional focus on text-to-image generation, we extend the application scope of our proposed image watermarking to the scenarios involving image-to-image diffusion model. Our method turns out both preserving visual quality of generated images and providing robustness against a wide range of adversarial attacks.

In summary, our paper contributes the following:

- We present a unified taxonomy that identifies and connects the key design principles underlying state-of-the-art watermarking identification methods.

- Under the unified watermark identification taxonomy, we propose a novel, training-free watermarking method, and theoretical analysis showing preservation of the latent representation distribution is accompanied.

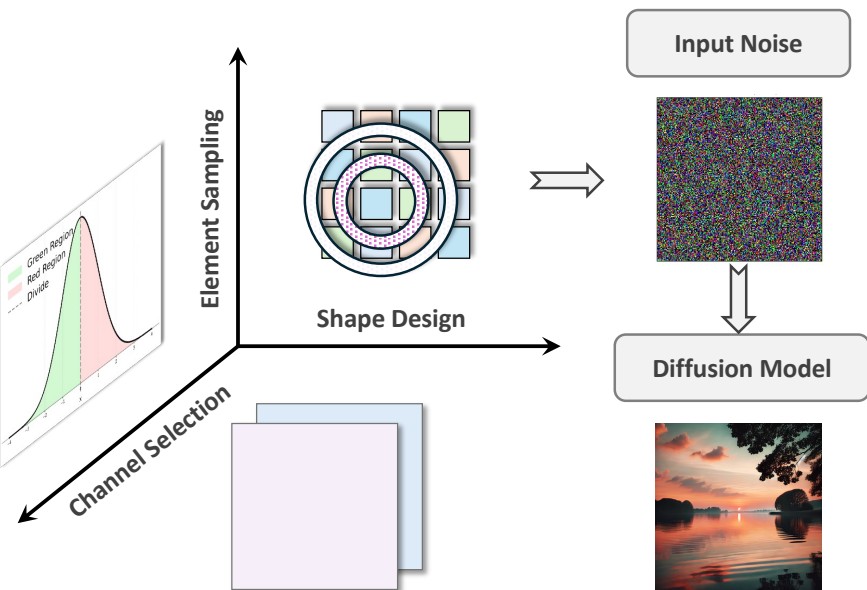

Figure 2: **Structured overview of diffusion watermarking.** In the embedding phase, watermark signals are injected into the latent noise along three orthogonal dimensions: (*i*) element sampling, which selects latent positions for modification; (*ii*) shape design, defining geometric structures such as rings or patches; and (*iii*) channel selection, identifying robust latent channels. The modified latent is then denoised by the diffusion model to produce the watermarked image.

- We extend watermarking techniques beyond the traditional focus on text-to-image generation scenario, and give the first systematic approach to watermarking image-to-image diffusion models. Our proposed method exhibits robustness against a variety of adversarial attacks and high visual quality, validating the efficacy of our method as well as the unified framework.

## 2 Preliminaries

In this section, we recap the diffusion process, LLM KGW watermarking, and provide a holistic framework for diffusion watermarking. A comprehensive survey of related work is provided in Section 6.

### 2.1 Diffusion Process

**Diffusion.** This section recaps key concepts of Latent Diffusion Models (LDMs), focusing on the diffusion process, denoising methods like DDIM, and inversion techniques for watermark detection. To map a regular image $x \in \mathbb{R}^{H \times W \times 3}$ to the latent space, Latent Diffusion Models (LDMs) formally behave as an autoencoder, using an encoder $E$ to obtain the representation $z_0$ as $z_0 = E(x) \in \mathbb{R}^{h \times w \times c}$; conversely, a decoder $D$ reconstructs the image $x$ from the latent space as $x = D(z_0)$. In generating images, (dropping the encoder $E$ and) directly feeding a random signal $z_0$ to the decoder $D$ of a *pre-trained* LDM can provably return an image following the pre-training distribution. In addition to the autoencoder framework, LDMs leverage a diffusion-like process to obtain $z_0$. Specifically, an initial latent representation $z_T \in \mathbb{R}^{h \times w \times c}$ is first sampled from a standard Gaussian distribution $\mathcal{N}(0, I)$; subsequently, iterative denoising methods like DDIM (Song et al., 2020) are used to transform $z_T$ into $z_0$, and the decoder then generates the image: $x = D(z_0)$.

**Inversion**. Beyond standard LDMs, **inversion techniques** (Dhariwal & Nichol, 2021) enable moving in the opposite direction—from a generated image back to the initial noise state. Previous empirical findings (Wen et al., 2023) suggest that DDIM inversion reliably reconstructs the initial noise, with $\hat{z}_T \approx z_T$. DDIM inversion can thus be used for watermark detection. Given a generated image $x$ and its associated starting noise $z_T$, we apply DDIM inversion to recover $\hat{z}_T$. This property allows us to compare the reconstructed noise to the original to detect embedded watermarks effectively.

## 2.2 LLM KGW Watermark

Here we introduce the basic LLM KGW watermark (Kirchenbauer et al., 2023; Zhao et al., 2024b) briefly. Let $s$ be a secret key, and let $t$ be a token from the vocabulary $\mathcal{V}$. We can partition $\mathcal{V}$ into two subsets of equal size: GreenList and RedList.

Given the original distribution $p(t \mid C)$ over tokens under context $C$, KGW watermarking modifies sampling by allowing tokens only from the GreenList:

$$
p_{\text{trunc}}(t \mid C) = \begin{cases} \dfrac{p(t \mid C)}{Z}, & t \in \text{GreenList}, \\ 0, & t \in \text{RedList}, \end{cases}
$$

where $Z = \sum_{t' \in \text{GreenList}} p(t' \mid C)$ is a normalizing constant. This ensures that all sampled tokens belong to the GreenList, leaving a distinct distributional footprint that can be verified during test time.

## 3 A Structured Overview of Watermarking

**Pipeline overview.** Figure 2 illustrates a concise end-to-end overview of our approach, covering both the embedding and detection phases of diffusion watermarking. To embed a robust and stealthy watermark, we categorize latent watermarking methods into three orthogonal design dimensions: ❶ **element sampling**, which selects latent positions for watermark injection; ❷ **watermark shapes**, defining geometric structures such as rings or patches; and ❸ **channel selection**, identifying latent channels that yield the most stable watermark signals. In the embedding phase, watermark bits are injected into the latent noise $z_T$ according to these dimensions, and the modified latent $\tilde{z}_T$ is then denoised by the diffusion model to produce the final watermarked image. During detection, DDIM inversion reconstructs the latent $\hat{z}_T$ from a given image (possibly attacked), and the same three dimensions are used to statistically verify watermark presence.

**Sampling of Individual Elements.** The first dimension concerns how values are assigned to each element. Tree-ring (Wen et al., 2023) sets a constant value in the Fourier frequency domain within each circle, forming a ring-like pattern. Ring-ID (Ci et al., 2024) refines this by applying the pattern only to the real part of the Fourier transform, enhancing imperceptibility. Post-hoc methods like DwtDctSVD (Cox et al., 2007) modify wavelet coefficients, adjusting DCT singular values in the DWT domain for watermark embedding. Gaussian-shading (Yang et al., 2024) embeds the watermark in the spatial domain by sampling values from a constrained distribution. Learning-based methods, such as Stable Signature (Fernandez et al., 2023) and AquaLoRA (Feng et al., 2024), tune model weights to integrate the watermark.

**Design of Watermark Shapes.** The second dimension defines the watermark shape within each $h \times w$ channel. Tree-ring (Wen et al., 2023) and Ring-ID (Ci et al., 2024) use concentric rings in the frequency domain to target specific components, improving robustness against geometric transformations. DwtDctSVD (Cox et al., 2007) embeds watermarks into pixel blocks within the combined DWT and DCT domains, while Gaussian-shading (Yang et al., 2024) also employs block-based embedding to improve noise resistance. Learning-based methods distribute the watermark across all locations, adapting the embedding to training conditions.

**Choice of Channels for Watermark Embedding.** The third dimension involves selecting channels for watermark embedding among the $c$ channels. Tree-ring (Wen et al., 2023) and post-hoc methods like DwtDctSVD (Cox et al., 2007) embed watermarks in specific channels based on empirical insights, prioritizing those less sensitive to perceptual changes to preserve image quality. In contrast, Gaussian-shading (Yang et al., 2024), Ring-ID (Ci et al., 2024), and learning-based methods distribute the watermark across all channels with empirical selection strategies, utilizing the full latent space to enhance robustness against attacks.

**Insights from the Unified Framework.** By structuring latent watermarking methods into three dimensions, our framework enables systematic comparison and reveals key challenges in existing approaches: how to sample individual elements akin to distribution-preserving methods in LLM watermarking for stealthy yet

detectable embedding, how to design watermark shapes resilient to both noise and geometric transformations, and how to develop channel selection for effective watermarking.

## 4 Methodology

In this section, based on the identified dimensions, we generalize the LLM KGW watermark to image diffusion models by preserving the distribution, as well as addressing vision-specific challenges in shape and channel dimensions to enhance the robustness and stealthiness of the watermark.

### 4.1 Basic: Generalizing LLM Watermarking

As discussed in Section 3, previous methods like Tree-ring, Ring-id, and DwtDctSvd heavily relied on setting selected watermarked elements to pre-defined constants. However, the fixed-value operations can substantially disrupt the IID nature of the latent representations, considering the $h \times w \times c$ elements in a latent representation are indeed IID standard univariate Gaussians $\mathcal{N}(0, 1)$. Moreover, under various attacks these elements can easily be altered, making the watermark difficult to detect.

To overcome these limitations, we propose a distribution-preserving watermarking technique (the characteristic of distribution preservation is reflected in Lemma 1). Inspired by the LLM KGW watermarking method (Kirchenbauer et al., 2023; Zhao et al., 2024b) widely studied in large language models (LLMs), we aim to adapt its principles to diffusion models. However, directly applying KGW to diffusion models is non-trivial. Unlike LLMs, which sample discrete tokens from a categorical distribution, diffusion models operate in a continuous latent space, requiring a distinct approach to maintain distribution preservation. To this end, we divide the density function $\phi(x)$ of the standard Gaussian distribution into two regions (Red and Green) of equal probability 0.5. Specifically, we partition the standard Gaussian distribution at the origin, yielding two **TruncatedGaussian** distributions over $(-\infty, 0]$ and $(0, \infty)$. Each latent element $z_T^e$ is assigned to one of these truncated intervals based on the watermark bit $m \in \{0, 1\}$, ensuring that the latent representations adhere to the following distribution:

$$p_{\text{trunc}}(z_T^e \mid m) = \begin{cases} \frac{\phi(z_T^e)}{\Phi(0)} & \text{if } m = 0, \quad z_T^e \leq 0 \\ \frac{\phi(z_T^e)}{1 - \Phi(0)} & \text{if } m = 1, \quad z_T^e > 0 \\ 0 & \text{otherwise (Red Region)} \end{cases} \tag{1}$$

where $\phi(\cdot)$ is the standard Gaussian density function, and $\Phi(\cdot)$ is its cumulative distribution function (CDF), where $\Phi(0) = 0.5$. To be specific, this distribution can be computed as follows:

$$z_T^e = \begin{cases} -|\xi|, & \text{if } m = 0, \\ +|\xi|, & \text{if } m = 1, \end{cases} \qquad \xi \sim \mathcal{N}(0, 1). \tag{2}$$

Figure 3 illustrates the sampling process of our proposed method. This approach can also be extended to multiple (i.e., more than 2) cumulative probability portions to encode more watermark bits.

The following lemma shows the proposed watermarking technique is distribution-preserving (a detailed proof and its generalized multi-bit extension are provided in Appendix B.1)

**Lemma 1** (Marginal Distribution of Elements (Galli et al., 1994; Del Castillo, 1994; Yang et al., 2024))**.** *Every element in the latent representation of truncated Gaussian marginally follows the standard normal distribution $\mathcal{N}(0, 1)$.*

### 4.2 Robustness and Stealthiness via Shape Design

Compared to LLM watermarking, image watermarking requires specifying the watermarking shape for each channel, i.e., the collection of elements to be watermarked; watermarks uniformly applied across the entire input can be overly sensitive to various transformations, making detection vulnerable to noise, distortions, or local perturbations. In this regard, previous works primarily define watermark embedding shapes as

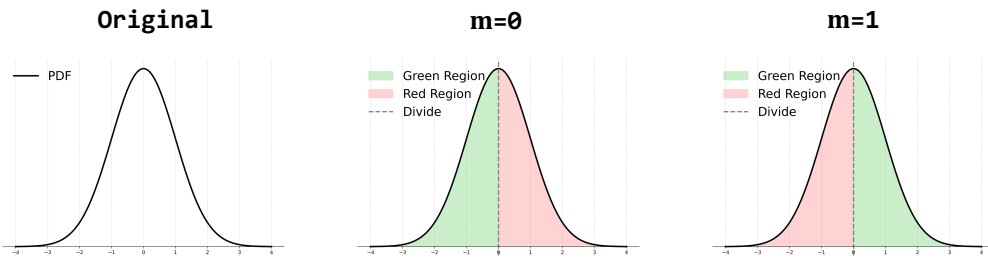

Figure 3: Standard normal distribution partitioning based on watermark value $m$. The left shows the original Probability Density Function (PDF), while the middle and right plots illustrate the "green" and the "red" regions for $m = 0$ and $m = 1$, respectively, with the dividing line at $x = 0$. The values are sampled from the green region.

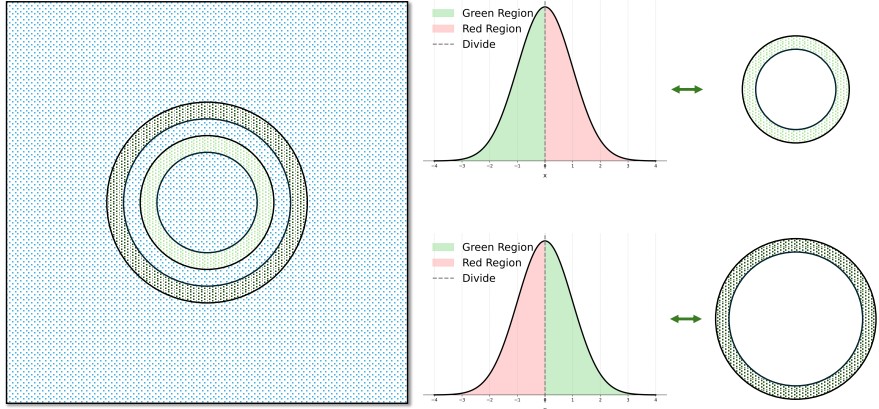

Figure 4: Visualization of Gaussian Ring Watermarking for geometric robustness. The left image shows two Gaussian rings embedded in the latent space, with deep green and light green indicating the two rings. The top and bottom examples on the right illustrate how different truncated Gaussian distribution are mapped to these rings in the latent space.

ring-shaped (Wen et al., 2023), block-shaped (Cox et al., 2007), or learnable values distributed across all locations (Zhang et al., 2019). In this section, we will illustrate how to specify the shapes for the watermark, to improve robustness and stealthiness.

**"Gaussian Ring" Watermarking for Geometric Robustness.** Given the more specific requirement to handle geometric transformations, we introduce an alternative approach to complement our framework: injecting the so-called "Gaussian Rings" into the representation of the latent space. The concept of "ring" implies a representation tensor is divided by a series of disjoint rings, which naturally inherits the "redundancy" idea and enhances robustness.

In detail, each Gaussian Ring is a meticulously structured ring-shaped watermark carrying a specific watermark value, designed to provide rotational invariance. As shown in Figure 4, the elements on a ring with a specific radius share the same watermark value, all sampled from a truncated Gaussian with a designated ("green") region. Our method embeds ring-shaped watermarks directly in the spatial domain, enabling seamless integration with the diffusion process without requiring frequency-based transformations. The process is illustrated in Algorithm 1 of Appendix A.2.

**"Random Gaussian" Watermarking with Redundant and Dispersed Watermarks.** The idea of "Random Gaussian" is to introduce redundancy to attain resistance to attacks. Inspired by Vision Transformers (Dosovitskiy et al., 2020), we split the initial input into patches, each carrying an identical watermark matrix $W$ of the same shape as the patch; every element of $W$ is a watermark value $m$ denoting the "green" region. This redundancy strengthens robustness, as information from multiple patches can be

aggregated during detection; even if some patches are compromised, ensemble methods help reconstruct the watermark, as formalized in the proposition in the next subsection.

However, redundant patterns over block-shaped patches notably introduces another challenge—the substantial degradation of generation quality due to the artificial structuring of the input. To maintain image fidelity while preserving watermark robustness, we move away from the concept of **fixed block-shaped** patches, and instead suggest **randomly coordinated** shapes that exhibit more natural distribution (c.f. the illustration in Figure 5 and the technical analysis in Proposition 2). Technically, we will uniformly sample a permutation of the representation elements and then adopt the block-shaped specification, which thoroughly **disperse** the elements. To ensure high-quality randomness, we employ the Chacha20 encryption algorithm (Bernstein et al., 2008). The method is illustrated in Algorithm 2 of Appendix A.2.

### 4.3 Theoretical Analysis on Robustness and Stealthiness

We first show that ensemble methods of multiple patches help ensure reliable and robust detection.

**Proposition 1** (Patch Aggregation Robustness). *Let $z_T$ represent the latent representation in a diffusion model, with $r$ denoting the probability of independently correctly detecting a watermark for an individual pixel. Given $p$ patches, each containing $n$ elements, assume $r > \frac{1}{2}$ and $p \gg 1$. Then, the probability $r_p$ of jointly detecting the watermark across all patches can be approximated by*

$$r_p \approx \Phi\left(\frac{(r - \frac{1}{2})\sqrt{p}}{\sqrt{r(1-r)}}\right).$$

*where $\Phi(x)$, the cumulative distribution function of a normal distribution, is a monotonically increasing function.*

We model the detection outcomes across $p$ patches as independent Bernoulli variables with parameter $r$. By invoking the Central Limit Theorem, the proportion of correctly detected patches can be approximated by a Gaussian distribution $\mathcal{N}\left(r, \frac{r(1-r)}{p}\right)$.

This formulation provides a key insight into the aggregation mechanism: the variance scales inversely with the number of patches. Consequently, provided that the single-patch detection is better than random guessing $(r > \frac{1}{2})$, the probability mass rapidly concentrates around the mean as $p$ increases, effectively separating the distribution from the random guessing baseline. This variance reduction thus boosts the joint detection probability $r_p$, ensuring robust performance even with weak individual signals, as evidenced in Table 4.

Stealthiness means that the watermark is visually imperceptible and that watermarked and non-watermarked images remain indistinguishable in distribution. In our setting, this is enforced by preserving the sampling distribution of the latents during watermark injection. To validate the claim that dispersed watermarks induce more natural representation distribution, we show in the following proposition that the correlation of two elements quickly diminish with the patch size $n$, and the limiting covariance matrix is thus akin to the one of a multivariate standard normal distribution. (The proof is provided in Appendix B.2.)

**Proposition 2** (Stealthiness of the Watermarked Distribution). *Let $p$ be the number of patches and $n$ indicate the number of pixels in a patch. The normalized Bures–Wasserstein (BW) distance $\hat{d}_{BW}$ between the watermarked sampling distribution and $\mathcal{N}(0, I)$ is at most*

$$\hat{d}_{BW}(z_T, \mathcal{N}(0, I)) \leq \frac{2}{\pi} \cdot \frac{p-1}{|z_T| - 1}.$$

*where $|z_T|$ denotes the size of $z_T$.*

The proposition above suggests that as long as $p \ll |z_T|$, the joint distribution of the watermarked latent representation $z_T$ well approximates the i.i.d. standard Gaussian distribution $\mathcal{N}(0, I)$ of the non-watermarked latents. Furthermore, by varying $p$, we can control the tradeoff between generation quality and robustness (also empirically validated in Table 4). For instance, in a latent representation of size $512 \times 512$, setting $p = 64$ results in a correlation of approximately $10^{-4}$, ensuring that the joint distribution closely follows an i.i.d. Gaussian.

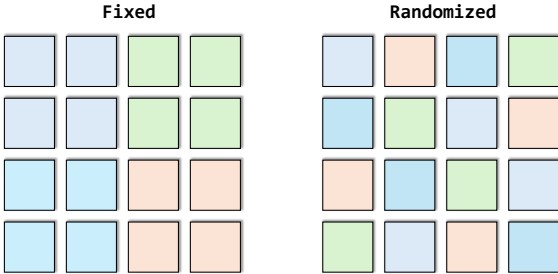

Figure 5: Comparison of Random Gaussian watermarking between fixed and randomized patches. Different colors represent distinct patches. The left plot shows a fixed patch arrangement, while the right plot displays a randomized patch configuration.

## 4.4 Channel Selection for Watermarking

To enhance robustness against various attacks, we propose a hybrid strategy that adaptively applies the "Random Gaussian" and "Gaussian Ring" techniques across channels, building on the shape specifications in Section 4.2.

**Imprinting.** To effectively integrate the two methods, we adopt a channel rating strategy to determine the watermarking technique to apply, specifically by calculating the sensitivity to geometric transformations. For a channel $c$ (a slight abuse of notation), we compute the magnitude of the gradient w.r.t. to a geometric transformation loss $\mathcal{L}_{\text{geo}}$:

$$g_c = \|\partial \mathcal{L}_{\text{geo}}/\partial z_T^c\|_2, \quad \mathcal{L}_{\text{geo}}(z_0, z_0^{\text{rot}}) = \|D(z_0) - D(z_0^{\text{rot}})\|_2^2,$$

where $z_T^c$ represents the initial latent representation of channel $c$, $D(z_0), D(z_0^{\text{rot}})$ denote the images generated by $z_T$ before and after 90 degree rotation, respectively, and $\|\cdot\|_2$ denotes the Euclidean norm. Clearly, the channel with the largest gradient magnitude $g_c$ is more sensitive to geometric transformations, and we will apply Gaussian Ring watermarking to enhance their robustness against geometric attacks. Conversely, for other channels with smaller gradient magnitudes, we will apply Random Gaussian watermarking to better handle non-geometric attacks. The full procedure is illustrated in Algorithm 3 of Appendix A.2.

**Detection aggregation along channels.** During the watermark detection process, we calculate the accuracy for each channel by evaluating whether the elements of $\hat{z}_T^{(c)}$ (a certain channel of the representation $\hat{z}_T$, an $h \times w$ matrix) fall into the specified "green" shape[1]. The so-called "accuracy" $\text{Acc}(z_T^{(c)}, w^c)$ for channel $c$ is calculated as:

$$\text{Acc}(\hat{z}_T^{(c)}, m^c) = \frac{1}{N_c} \sum_{i=1}^{N_c} \mathbf{1}\{z_i^c \in \text{Green Region}(m^c)\}$$

where $N_c$ is the number of watermarked elements in channel $c$, $z_i^c$ is the $i$-th element of channel $c$, $m^c$ is the watermark value for channel $c$, and $\mathbf{1}\{\cdot\}$ is the indicator function, which equals 1 if the event is true and 0 otherwise. The overall watermark recovery accuracy is determined by aggregating the accuracy across all channels. To adaptively address varying attack scenarios, we propose the combined watermark accuracy to select the most robust component:

$$\text{Acc}(\hat{m}) = \max_{c \in C_m} \text{Acc}(\hat{z}_T^{(c)}, m^c)$$

where $C_m$ is the set of all channels. By detecting both random and ring watermark patterns, even if one of the watermarks is compromised by a specific attack, the others can still be detected, enhancing overall robustness. By combining these methods, the total watermark capacity is greatly increased, as the combined capacity becomes the product of the individual capacities of each watermarking technique.

---

[1]In addition to the practical detection based on the accuracy metric, a testing procedure is depicted in Appendix B.3.

Table 1: TPR@1%FPR under different attacks for Stable Diffusion, showing the effectiveness of our method over a number of attacks.

| Method | Clean | Rotation | JPEG | Cr. & Sc. | Blurring | GauNoise | Color Jitter | S&PNoise | Regeneration | Flip | Avg |
|---|---|---|---|---|---|---|---|---|---|---|---|
| DwtDct | 0.909 | 0.027 | 0.008 | 0.092 | 0.011 | 0.354 | 0.126 | 0.089 | 0.016 | 0.023 | 0.166 |
| DwtDctSvd | 1.000 | 0.011 | 0.156 | 0.057 | 0.538 | 0.732 | 0.117 | 0.021 | 0.018 | 0.042 | 0.269 |
| RivaGan | 0.997 | 0.012 | 0.756 | 0.762 | 0.428 | 0.541 | 0.694 | 0.477 | 0.025 | 0.027 | 0.472 |
| Stable Signature | 1.000 | 0.032 | 0.713 | 0.816 | 0.015 | 0.624 | 0.843 | 0.072 | 0.011 | 0.018 | 0.414 |
| Tree-Ring | 1.000 | 0.477 | 0.995 | 0.932 | 0.999 | 0.926 | 0.900 | 0.987 | 1.000 | 1.000 | 0.922 |
| Gaussian Shading | 1.000 | 0.007 | 0.999 | 1.000 | 1.000 | 0.999 | 0.992 | 0.999 | 1.000 | 0.097 | 0.809 |
| AquaLoRA | 1.000 | 0.013 | 0.987 | 0.941 | 1.000 | 0.954 | 0.847 | 0.693 | 0.812 | 0.133 | 0.738 |
| **DiffKGW (Ours)** | **1.000** | **0.852** | **1.000** | **1.000** | **1.000** | **0.996** | **0.996** | **1.000** | **1.000** | **0.998** | **0.984** |

## 5 Empirical Results

We conduct experiments on two widely-used diffusion scenarios: text-to-image diffusion models and image-to-image diffusion models, to evaluate the effectiveness and robustness of our watermarking technique across various attack scenarios. Additionally, we perform ablation studies in Section 5.3 for a deeper analysis of the method.

### 5.1 Experimental Setting

Our paper first evaluates the proposed method on text-to-image diffusion models, focusing on Stable Diffusion (SD) (Rombach et al., 2022) v2.1/2.0/1.4. The generated images have a resolution of $512 \times 512$, with a latent space of $64 \times 64 \times 4$. For inference, we use prompts from Stable-Diffusion-Prompt, setting the guidance scale to 7.5, and generate images over 50 steps using DPMSolver (Lu et al., 2022). The watermark radius $r$ varies from 5 to 15 in steps of 2. Images are divided into patches of 64 elements each. Given that users often share generated images without prompts, we perform inversions with an empty prompt and a guidance scale of 1, using 50 steps of the DDIM method (Song et al., 2020).

We also test our watermarking approach in image-to-image editing using pre-trained image-conditioned diffusion models. Specifically, we employ instruct-pix2pix (Brooks et al., 2023), a fine-tuned Stable Diffusion model. Editing tasks use DDIM inversion with an empty prompt and original image, setting the guidance scale to 1.

For image quality evaluation, we adopt Frechet Inception Distance (FID) (Heusel et al., 2017) and CLIP-Score (Radford et al., 2021). FID is computed on the `COCO2017` validation set, which contains 5,000 images. CLIP scores are measured between generated images and text prompts for Stable Diffusion, and between edited images and ground truth descriptions for instruct-pix2pix. We exclude SSIM and PSNR, as they assess post-hoc modifications, whereas our method manipulates latent representations, aligning with prior works (Wen et al., 2023; Yang et al., 2024).

For detection, following (Wen et al., 2023; Yang et al., 2024), we compute the **true positive rate** (TPR) at a fixed **false positive rate** (1% FPR). In traceability, we assess identification accuracy across watermark patterns. AUC and TPR@1%FPR are calculated using 1,000 watermarked and 1,000 unwatermarked images per run, averaging results over three runs with different random seeds.

### 5.2 Performance of the Proposed Method

To benchmark the robustness of our watermarking method, we evaluate its performance against widely used augmentation-based attacks and a generative AI-based attack, Regeneration (Zhao et al., 2023a), which leverages a diffusion model for regeneration and denoising. We exclude image-to-image translation and editing methods, as they produce significantly altered images with varying concepts. Additional potential adversarial threats exist, such as recovering structured watermark patterns from degenerate or lightly edited images, reusing extracted watermark signals across samples, and overwriting an existing watermark with a new one. These attacks fall outside the scope of our current benchmark but represent meaningful directions for future investigation. Details of the attacks are in Appendix A.1, and Table 1 presents the TPR@1%FPR results. Our DiffKGW watermarking method outperforms Tree-Ring, particularly under rotation (0.852

Table 2: Performance comparison of different watermarking methods on Stable Diffusion under clean and adversarial conditions. The metrics are split into two categories: Fidelity and Robustness.

| Methods | Robustness | | | | | | Fidelity | |
|---|---|---|---|---|---|---|---|---|
| | TPR @1%FPR | | AUC | | Accuracy | | FID ($\downarrow$) | CLIP-Score |
| | Clean | Adv. | Clean | Adv. | Clean | Adv. | | |
| Stable Diffusion | - | - | - | - | - | - | 25.23 | 0.363 |
| DwtDct | 0.909 | 0.166 | 0.974 | 0.574 | 0.950 | 0.527 | 25.28 | 0.364 |
| DwtDctSvd | 1.000 | 0.269 | 1.000 | 0.702 | 1.000 | 0.691 | 25.01 | 0.359 |
| RivaGAN | 0.997 | 0.472 | 0.999 | 0.854 | 0.998 | 0.802 | **24.51** | 0.361 |
| Stable Signature | 1.000 | 0.414 | 1.000 | 0.818 | 1.000 | 0.751 | 25.45 | **0.364** |
| Tree-Ring | 1.000 | 0.922 | 1.000 | 0.993 | 1.000 | 0.979 | 25.29 | 0.363 |
| Gaussian Shading | 1.000 | 0.809 | 1.000 | 0.911 | 1.000 | 0.864 | 25.20 | 0.364 |
| AquaLoRA | 1.000 | 0.738 | 1.000 | 0.871 | 1.000 | 0.817 | 25.50 | 0.363 |
| **DiffKGW (Ours)** | **1.000** | **0.984** | **1.000** | **0.999** | **1.000** | **0.994** | 25.20 | 0.363 |

Table 3: Detection TPR with different sampling methods in diffusion models.

| Noise | Sampling Method | | | | |
|---|---|---|---|---|---|
| | DDIM | UniPC | PNDM | DEIS | DPMSolver |
| Clean | 1.000 | 1.000 | 1.000 | 1.000 | 1.000 |
| Adversarial | 0.978 | 0.972 | 0.983 | 0.982 | 0.984 |
| CLIP-Score | 0.363 | 0.362 | 0.363 | 0.363 | 0.363 |

vs. 0.477) and noisy conditions like Gaussian noise (0.996 vs. 0.926). While Gaussian Shading excels in noise robustness (e.g., 1.000 for blurring), it struggles with geometric transformations (0.007 for rotation). Overall, our method achieves the highest average performance, demonstrating strong resilience against both geometric and noise-based attacks. Additional results on the instruct-pix2pix model and image-to-image editing scenarios are in Table 6 (Appendix A.3), while watermark capacity and identification analyses are in Appendix A.4, further confirming our method's superiority.

Table 2 shows the average detection performance across various attacks, demonstrating our method's robustness and superiority over baselines in resisting adversarial manipulations. In fidelity, it maintains comparable FID and CLIP-Score values, ensuring minimal impact on image quality. Moreover, its independence from custom model components enables seamless integration into different diffusion models. Appendix A.7 presents results on SD v1.4, v2.0, and Dreambooth (Ruiz et al., 2023), further validating its effectiveness.

Our study focuses on watermarking methods that are *integrated into the diffusion generation process*, where watermark signals are embedded directly into the latent noise prior to denoising. This paradigm fundamentally differs from *post-processing* watermarking techniques that modify already-generated images (Lu et al.) and from optimization-based approaches (Zhang et al., 2024). Accordingly, we do not include direct comparisons.

## 5.3 Ablation Studies

We conduct extensive ablation studies on several key hyperparameters of our proposed method to demonstrate the effectiveness of our method. To validate the generalization of our approach, we evaluated five commonly used sampling methods in diffusion: DDIM (Song et al., 2020), UniPC (Zhao et al., 2024a), PNDM (Liu et al., 2022), DEIS (Zhang & Chen, 2022), and DPMSolver (Lu et al., 2022). As shown in Table 3, with our proposed watermarking technique, all sampling methods demonstrate excellent and comparable performance, particularly in clean conditions where all methods achieved a perfect detection rate. Under adversarial noise, DPMSolver shows a marginally better detection rate, but overall, all sampling methods maintain high robustness.

Table 4 shows that larger patch sizes improve robustness against adversarial attacks but reduce image quality, as reflected by a lower CLIP-Score. For ring radius, placing the ring near the center ("0-5" in Table 5) degrades generation quality by embedding key structural and semantic information, while a medium radius ("5-15") provides the best balance between rotation robustness and image quality.

Table 4: Ablation with different patch sizes.

| Patch Size | 4 | 16 | 64 | 256 |
|---|---|---|---|---|
| None | 0.993 | 1.000 | 1.000 | 1.000 |
| Adversarial | 0.757 | 0.924 | 0.984 | 0.986 |
| CLIP-Score | 0.364 | 0.363 | 0.363 | 0.359 |

Table 5: Ablation with different ring radii.

| Ring Radius | 0-5 | 5-10 | 0-10 | 5-15 | 10-15 |
|---|---|---|---|---|---|
| None | 1.000 | 1.000 | 1.000 | 1.000 | 1.000 |
| Rotation | 0.512 | 0.643 | 0.828 | 0.852 | 0.767 |
| CLIP-Score | 0.361 | 0.363 | 0.359 | 0.363 | 0.363 |

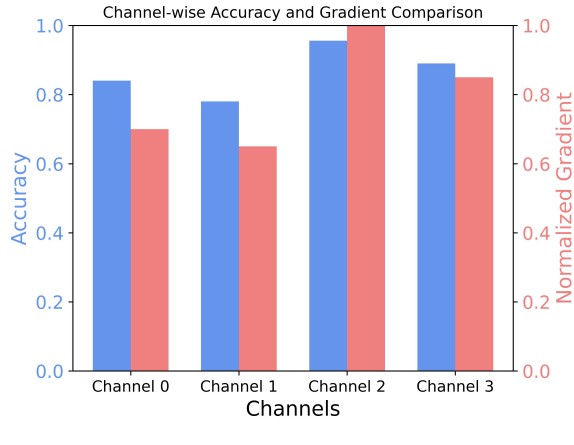

(a) Channel-wise analysis.

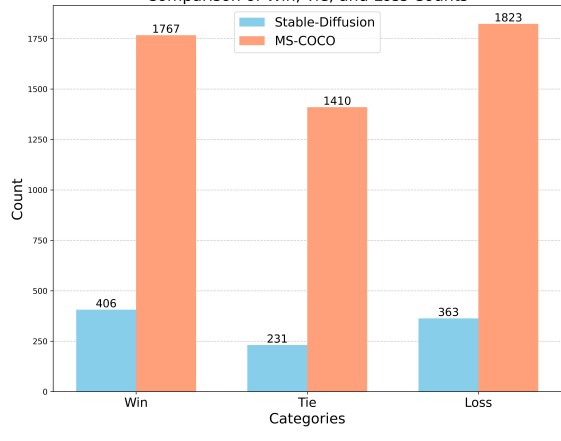

(b) Win/Tie/Loss comparison.

Figure 6: **Left:** Quantitative comparison of accuracy and normalized gradient across four channels. **Right:** Win/Tie/Loss statistics between watermarked and non-watermarked outputs on the Stable-Diffusion and MS-COCO datasets. "Win" indicates higher performance for watermarked images, "Tie" similar performance, and "Loss" higher for non-watermarked.

For watermark channels, the experimental findings in Figure 6a show substantial variation in performance across different channels, with Channel 2 achieving the highest accuracy. Notably, the computed gradient values align closely with the robustness accuracy for each channel, indicating that gradient strength is a reliable indicator of channel performance. This suggests that channels with stronger gradients are better suited for embedding the Gaussian Ring Watermark, providing a useful guideline for selecting the optimal channel for watermarking to enhance specific robustness.

Due to space constraints, ablations on Gaussian Ring and Random Gaussian watermarks are in Appendix A.5, and analyses on inversion steps, inference steps, and multi-bit encoding are in Appendix A.6, all demonstrating the effectiveness of our method.

### 5.4 Analysis on Channels

Additionally, we compare the CLIP-Scores of the non-watermarked and watermarked results for each input prompt from the Stable-Diffusion and MS-COCO datasets. If the CLIP-Score difference is less than 0.01, it is considered a tie; otherwise, it is classified as a win for either the watermarked or non-watermarked result. The comparative results are presented in Figure 6b.

From the table, we observe that the watermarked images sometimes outperform and other times underperform the non-watermarked images, with no consistent trend favoring one over the other (similar win and loss counts with win rate≈0.5). This demonstrates the robustness and effectiveness of our watermarking method, as it preserves the generation quality while embedding the watermark without introducing systematic degradation or enhancement.

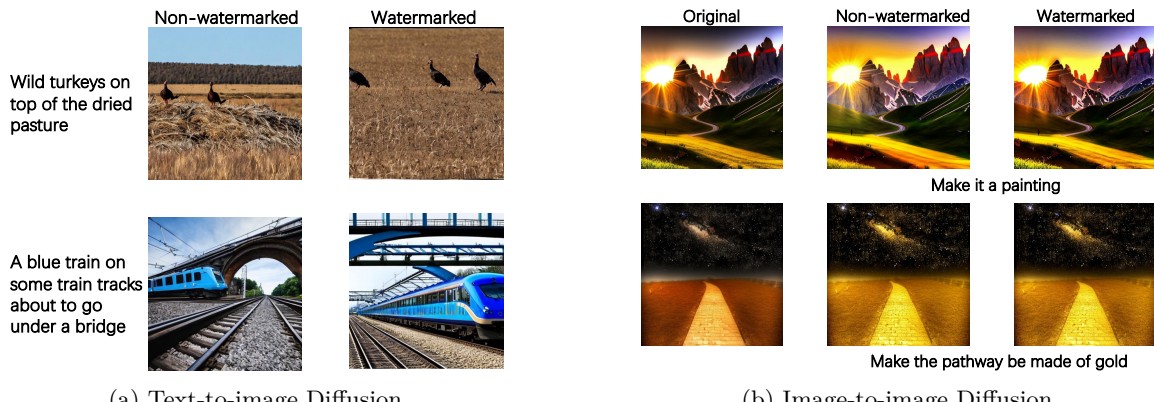

(a) Text-to-image Diffusion                    (b) Image-to-image Diffusion

Figure 7: Visualization of both the watermarked and non-watermarked generated images in different scenarios. For image-to-image editing, we also include the original images. More visualizations are shown in Appendix C.

## 5.5 Visualization

Figure 7 visualizes watermarked and non-watermarked diffusion-generated images, showing that our method preserves semantic information and high visual quality. In text-to-image diffusion, watermarked images remain visually similar to non-watermarked ones while retaining intended semantics. In image-to-image diffusion, they exhibit even greater similarity to the original and non-watermarked images due to additional guidance from the input. These results confirm that our method effectively embeds the watermark while maintaining image integrity. Additional visualizations are in Appendix C.

## 6 Related Works

In this section, we comprehensively review the related works on watermarking images and recent extensions to diffusion models.

**Watermarking Image.**  Digital watermarking (Van Schyndel et al., 1994) embeds traceable identification information in carrier data for copyright protection and content authentication. Traditional image watermarking techniques, often applied in post-processing, focus on frequency domain methods (Cox et al., 2007; Al-Haj, 2007; Hamidi et al., 2018; Kundur & Hatzinakos, 1997; Lee et al., 2007; Navas et al., 2008) to enhance robustness. For instance, DwtDctSvd (Cox et al., 2007) combines Discrete Wavelet Transform, Discrete Cosine Transform, and Singular Value Decomposition for watermark embedding. More recently, deep learning-based approaches (Zhu et al., 2018; Tancik et al., 2020; Fernandez et al., 2022; Zhang et al., 2019; Hayes & Danezis, 2017) like RivaGAN (Zhang et al., 2019) leverage neural networks to improve watermarking, employing adversarial networks for both embedding and extraction. Despite advancements, post-hoc watermarking often introduces visible noise and is vulnerable to attacks like cropping and compression (Fernandez et al., 2023), as it is applied to the final image, making the watermark prone to distortion or removal.

**Watermarking Diffusion Models.**  With the rise of generative models, particularly diffusion models, watermarking AI-generated content—or the models themselves—has become increasingly important. Yu et al. (2021) and Zhao et al. (2023b) proposed embedding watermarks into training datasets so that models inherently generate watermarked content. However, this approach may pose practical issues for large-scale diffusion models trained on vast datasets. To address this, researchers have explored embedding watermarks during the generation process. For example, Fernandez et al. (2023) and Feng et al. (2024) fine-tuned model weights to modify latent representations, with Stable Signature (Fernandez et al., 2023) fine-tuning LDM decoders to embed hidden watermarks in generated images—albeit at the cost of extensive training and various data augmentations.

Other approaches, like Tree-ring watermark (Wen et al., 2023), modify the initial noise in the sampling process, requiring deterministic samplers like DDIM (Song et al., 2020) for watermark extraction through

inversion. Yang et al. (2024) proposes a distribution preserving method to adjusts the initial noise naturally, improving robustness against noise additions. However, Wen et al. (2023) is less resilient to noise and cropping attacks, while Yang et al. (2024) remains sensitive to geometric transformations. Despite these efforts, the fundamental principles behind effective watermarking and the connections among different methods remain poorly understood. Inspired by these works, we identify key design dimensions for embedding watermarks in the latent space, providing insights into diffusion watermarking techniques.

## 7 Conclusion

In this paper, we adapt the principle of language watermarking to image diffusion and introduce a unified framework to dissect watermarking approaches for diffusion models along three distinct dimensions. By addressing vision-specific challenges in shape and channel dimensions, we innovatively instantiate an effective holistic model under this framework, maintaining high fidelity, and ensuring robust watermarking against various attacks. Extensive evaluations of text-to-image applications show our model outperforms state-of-the-art methods. We further validate its effectiveness on image-to-image diffusion models, highlighting a substantial advancement in digital watermarking.

## Acknowledge

This work is supported by National Science Foundation under Award No. IIS-2117902. The views and conclusions are those of the authors and should not be interpreted as representing the official policies of the funding agencies or the government.

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

# Appendix

# Contents

# A  More on experiments

The code and model weights will be open-sourced after the review procedure.

## A.1  Implementation Details

The attacks implemented in the paper include Gaussian blurring with a filter size of 4, Gaussian noise with a standard deviation of 0.05, JPEG compression with a ratio of 25%, and salt-and-pepper noise with a probability of 0.05. Additionally, we evaluate against brightness adjustments where the factor is set to 6, random cropping and resizing with a 75% ratio, rotation by 75 degrees, and horizontal and vertical flips with probabilities of 0.5 each. We also incorporate a generative model-based attack, DeNoise, as introduced by Zhao et al. (2023a), which leverages a diffusion model to denoise the output, aiming to remove any potential watermark. To be specific, their method first encodes the watermarked image into the model's latent space, then applies 50 forward diffusion steps, consistent with our diffusion setup, to obtain a noised latent representation. It subsequently runs the reverse denoising trajectory to recover a cleaner latent, which is finally decoded back into pixel space to produce an image intended to suppress or eliminate the embedded watermark.

For the Gaussian Ring Watermark, the watermark radius $r$ ranges from 5 to 15, with an interval of 2. For the Random Gaussian Watermark, we divide the images into patches, with each patch containing 64 elements. Furthermore, to enhance the randomness and security of the watermark, we employ a stream key, a cryptographic key used in encryption algorithms to generate a sequence of pseudo-random values. Following Yang et al. (2024), we employ a stream key to encrypt the watermark, which consists of binary values (0/1), into a randomized version **m** using an encryption method like ChaCha20 (Bernstein et al., 2008). The encrypted watermark $W$, now uniformly distributed, avoids detectable artifacts while maintaining alignment with the natural data distribution.

## A.2  Pseudo Code of Proposed DiffKGW

In this subsection, we illustrate the pseudo-code of the algorithms in our proposed DiffKGW framework.

---
**Algorithm 1** Gaussian Ring Watermarking (Channel $c$)

---
**Require:** $z_T^{(c)} \in \mathbb{R}^{h \times w}$, ring boundaries $\{r_0, r_1, \ldots, r_K\}$, pre-randomized watermark bits $\{b_1, b_2, \ldots, b_K\}$
1: **for** each pixel $(i, j)$ in $z_T^{(c)}$ **do**
2:     dist $\leftarrow \sqrt{(i - h/2)^2 + (j - w/2)^2}$
3:     **for** $k = 1$ **to** $K$ **do**
4:         **if** $r_{k-1} \leq \text{dist} < r_k$ **then**
5:             $m \leftarrow b_k$                                     ▷ Retrieve watermark bit for ring $k$
6:             $z_T^{(c)}(i, j) \leftarrow \text{TruncatedGaussian}(m)$ Eq. (1)
7:         **end if**
8:     **end for**
9: **end for**
10: **return** $z_T^{(c)}$

---

## A.3  Empirical Results on Instruct-pix2pix

The experimental findings in Table 6 demonstrate that, even in the more challenging image-to-image diffusion scenarios, our method maintains high robustness across various attack types, with an average AUC of 0.927. In contrast, the performance of the Tree-Ring method significantly declines, possibly because it sets selected values to constants, heavily relying on inversion precision. In image-to-image settings, inversion is more difficult due to the absence of multiple inputs including the conditional editing prompt and input image during the inversion process. This highlights the strength of our method, which can achieve robustness even when inversion accuracy is compromised, further underscoring its effectiveness in complex scenarios.

---

**Algorithm 2** Gaussian Random Watermarking (Channel $c$)

---

**Require:** $z_T^{(c)} \in \mathbb{R}^{h \times w}$, patch size $p$, pre-randomized watermark matrix $W \in \{0,1\}^{(h/p) \times (w/p)}$

1: Partition $z_T^{(c)}$ into patches of size $p \times p$
2: **for** each patch $k$ **do**
3:   **for** each pixel $(i_p, j_p)$ in the corresponding patch **do**
4:     $m \leftarrow W(i_p, j_p)$                                    ▷ Retrieve watermark bit
5:     $P^{(k)}(i_p, j_p) \leftarrow \text{TruncatedGaussian}(m)$ Eq. (1)
6:   **end for**
7: **end for**
8: Combine all patches $\{P^{(1)}, P^{(2)}, \ldots, P^{(p \times p)}\}$ to reconstruct $z_T^{(c)}$
9: **return** $z_T^{(c)}$

---

**Algorithm 3** DiffKGW

---

**Require:** $z_T \in \mathbb{R}^{h \times w \times C}$, Decoder $D(\cdot)$, Rotation function $\text{rot}(\cdot)$, $L_{geo}(z_0, z_{\text{rot0}}) = \|D(z_0) - D(z_{\text{rot0}})\|_2^2$

1: **for** $c = 1$ to $C$ **do**
2:   Extract channel $z_T^{(c)}$
3:   $I \leftarrow D(z_T)$, $I_{\text{rot}} \leftarrow D(\text{rot}(z_T))$
4:   Compute gradient $g_c \leftarrow \left\| \dfrac{\partial L_{geo}}{\partial z_T^{(c)}} \right\|_2$
5: **end for**
6: $c^* \leftarrow \arg\max_c g_c$
7: **for** $c = 1$ to $C$ **do**
8:   **if** $c = c^*$ **then**
9:     Apply **Gaussian Ring** (Alg. 1) on $z_T^{(c)}$
10:    **else**
11:     Apply **Gaussian Random** (Alg. 2) on $z_T^{(c)}$
12:    **end if**
13: **end for**
14: **return** Watermarked latent $\hat{z}_T$

---

### A.4 Watermark Capacity and Identification

The experimental results in Table 7 demonstrate the superior capacity and robustness of our watermarking method compared to the Tree-Ring approach. In our method, two distinct watermarks are injected into different channels, each carrying $a$ and $b$ bits of information. This results in $2^a$ and $2^b$ possible patterns for each channel, and when combined, the total capacity becomes $2^{a+b}$. This significantly increases the capacity and enhances the distinguishability of the watermarks compared to constant value-based approaches like Tree-Ring. The results show that across various use cases and watermark patterns, our method consistently achieves high identification accuracy, while the Tree-Ring approach fails, especially under adversarial conditions. These findings underscore the effectiveness of our method in maintaining both traceability and identification accuracy, even in challenging adversarial scenarios.

### A.5 Ablation Study on Random Gaussian and Gaussian Ring

The experimental results in the Table 8 show that both the Random Gaussian and Gaussian Ring watermark patterns are critical to the success of our method. The Random Gaussian watermark demonstrates greater robustness against noise-based attacks, as indicated by the high performance in noise-related tests (e.g., Salt & Pepper Noise: 0.996), while the Gaussian Ring watermark shows superior robustness to geometric transformations, such as rotation (0.011 for without Gaussian Ring, compared to 0.841 with Gaussian Ring). Despite this, both patterns perform well individually, achieving strong results across various attack scenarios. The combination of the two watermarks in our method results in the highest overall performance (average

Table 6: AUC under each Attack for Instruct-pix2pix image-to-image diffusion, showing the effectiveness of our method over a number of attacks.

| Method | Clean | Rotation | JPEG | Cr. & Sc. | Blurring | GauNoise | Color Jitter | S&PNoise | DeNoise | Flip | Avg |
|---|---|---|---|---|---|---|---|---|---|---|---|
| DwtDct | 0.542 | 0.301 | 0.322 | 0.288 | 0.315 | 0.274 | 0.265 | 0.292 | 0.268 | 0.279 | 0.315 |
| DwtDctSvd | 0.603 | 0.312 | 0.411 | 0.355 | 0.428 | 0.362 | 0.344 | 0.305 | 0.318 | 0.293 | 0.373 |
| RivaGan | 0.712 | 0.366 | 0.557 | 0.521 | 0.498 | 0.476 | 0.505 | 0.422 | 0.388 | 0.351 | 0.480 |
| Stable Signature | 0.734 | 0.401 | 0.612 | 0.586 | 0.445 | 0.512 | 0.587 | 0.355 | 0.392 | 0.336 | 0.496 |
| Tree-Ring | 0.751 | 0.586 | 0.689 | 0.653 | 0.695 | 0.625 | 0.615 | 0.673 | 0.602 | 0.605 | 0.649 |
| Gaussian Shading | 0.963 | 0.532 | 0.961 | 0.924 | 0.943 | 0.912 | 0.942 | 0.912 | 0.916 | 0.525 | 0.853 |
| AquaLoRA | 0.781 | 0.452 | 0.703 | 0.631 | 0.702 | 0.655 | 0.598 | 0.474 | 0.501 | 0.402 | 0.590 |
| **Ours** | **0.988** | **0.830** | **0.976** | **0.947** | **0.970** | **0.914** | **0.946** | **0.920** | **0.942** | **0.841** | **0.927** |

Table 7: Traceability and identification accuracy across 32 distinct watermark patterns.

| Method | Clean | Rotation | JPEG | Cr. & Sc. | Color Jitter | GauNoise | Avg |
|---|---|---|---|---|---|---|---|
| Tree-Ring | 0.435 | 0.012 | 0.401 | 0.045 | 0.412 | 0.505 | 0.302 |
| **Ours** | **1.000** | **0.828** | **0.992** | **0.984** | **0.980** | **0.982** | **0.961** |

0.984), demonstrating that these patterns are not only effective but also complementary, enhancing the overall robustness when used together.

## A.6 Ablation Study on Inference and Inversion Steps.

The experimental findings in Table 10 demonstrate that different inversion steps consistently perform well in terms of detection accuracy, with minimal loss even when there is a mismatch between inference and inversion steps. In real-world scenarios, the exact inference step is often unknown, which can result in this mismatch. However, the table shows that detection performance remains robust across various combinations of steps. Given the efficiency of existing samplers and the optimal performance observed with 50 inversion steps, we select 50 steps as a balanced trade-off between accuracy and computational efficiency.

Encoding multiple $k$ bits involves dividing the distribution into $2^k$ regions, designating one as the green region while treating all others as red. The detection performance remains comparable, with an increase in encoded bits generally enhancing detection. However, encoding too many bits per element may may negatively impact overall image quality.

## A.7 Performance on Different Diffusion Models

Our method is independent of custom model components, allowing seamless integration into various diffusion models (DMs). In the paper, we evaluate our method on InstructPix2Pix and Stable Diffusion (SD) V2.1. Additionally, we further validate its generalizability on SD V1.4, V2.0, and DreamBooth below, covering different architectures, training data, and objectives. Our method shows consistent robustness.

## A.8 Runtime Analysis

As shown in Table 11, the actual watermarking cost is minimal: Tree-Ring is the fastest, but our method also requires only 0.028 s for embedding and 0.0017 s for detection. In contrast, diffusion and inversion take over a second, meaning our watermark introduces minimal overhead to the generation pipeline.

## A.9 Robustness Evaluation

As shown in Figure 8, increasing the intensity of blur or noise progressively reduces the watermark detection rate, which is expected because strong corruptions severely distort the image itself. Even under extremely destructive conditions such as when 80% of the pixels are replaced by noise, our method still maintains a

Table 8: Comparison of detection performance (TPR@1%FPR) across various methods under different attack scenarios, including Random Gaussian removal, Gaussian Ring removal, and the proposed method (Ours).

| Method | Clean | Rotation | JPEG | Cr. & Sc. | Blurring | GauNoise | Color Jitter | S&PNoise | DeNoise | Flip | Avg |
|---|---|---|---|---|---|---|---|---|---|---|---|
| w/o Random Gaussian | 1.000 | 0.841 | 0.966 | 0.954 | 0.937 | 0.923 | 0.982 | 0.984 | 0.931 | 0.997 | 0.952 |
| w/o Gaussian Ring | 1.000 | 0.011 | 1.000 | 1.000 | 0.986 | 0.994 | 0.988 | 0.996 | 1.000 | 0.015 | 0.799 |
| **Ours** | **1.000** | **0.852** | **1.000** | **1.000** | **1.000** | **0.996** | **0.996** | **1.000** | **1.000** | **0.998** | **0.984** |

Table 9: Detection TPR@1%FPR with different inversion and inference steps.

| | Inversion Step | | | |
|---|---|---|---|---|
| **Inference Step** | **10** | **25** | **50** | **100** |
| 10 | 0.975 | 0.976 | 0.973 | 0.970 |
| 25 | 0.968 | 0.978 | 0.981 | 0.981 |
| 50 | 0.965 | 0.967 | 0.984 | 0.982 |
| 100 | 0.965 | 0.966 | 0.977 | 0.984 |

reasonably high detection accuracy. This demonstrates that the watermark signal remains resilient even when the visual content is heavily degraded.

# B  Proofs omitted in the main text

## B.1  Proof of Lemma 1

*Proof.* Let $Z$ be a random variable representing the value of an element in the latent representation. The pixel $Z$ is sampled based on a watermark bit $B \in \{0, 1\}$, which determines the region of the Gaussian distribution from which $Z$ is drawn.

The conditional distribution of $Z$ given $B$ is:

$$p_{Z|B}(z_T^e|w) = \begin{cases} 2\phi(z_T^e), & \text{if } z_T^e \in R(w), \\ 0, & \text{otherwise,} \end{cases}$$

where $\phi(x) = \dfrac{1}{\sqrt{2\pi}}e^{-x^2/2}$ is the standard normal probability density function, and $R(w)$ is defined as:

$$R(w) = \begin{cases} (-\infty, 0], & \text{if } w = 0, \\ (0, \infty), & \text{if } w = 1. \end{cases}$$

The watermark bits $B$ are assumed to be independent and uniformly random, i.e., $P(B = 0) = P(B = 1) = \dfrac{1}{2}$. The marginal distribution of $Z$ is then:

$$\begin{aligned} p_Z(z_T^e) &= \sum_{w \in \{0,1\}} p_{Z|B}(z_T^e|w)P(B = w) \\ &= \frac{1}{2} \cdot 2\phi(z_T^e)\mathbf{1}_{z_T^e \le 0} + \frac{1}{2} \cdot 2\phi(z_T^e)\mathbf{1}_{z_T^e > 0} \\ &= \phi(z_T^e)\left[\mathbf{1}_{z_T^e \le 0} + \mathbf{1}_{z_T^e > 0}\right] \\ &= \phi(z_T^e). \end{aligned}$$

Therefore, $Z$ marginally follows the standard normal distribution $\mathcal{N}(0, 1)$. □

Table 10: Detection TPR@1%FPR with different encoding watermark bits per element.

| # Bits | 1 | 2 | 3 |
|---|---|---|---|
| None | 1.000 | 1.000 | 1.000 |
| Adversarial | 0.984 | 0.985 | 0.987 |
| CLIP-Score | 0.363 | 0.363 | 0.360 |

| Model | SD v1.4 | | | SD v2.0 | | | DreamBooth | | |
|---|---|---|---|---|---|---|---|---|---|
| | Clean | Adv. | CLIP-Score | Clean | Adv. | CLIP-Score | Clean | Adv. | CLIP-Score |
| Base Model | - | - | 0.349 | - | - | 0.358 | - | - | 0.352 |
| +Gaussian Shading | 1.000 | 0.788 | 0.348 | 1.000 | 0.814 | 0.358 | 1.000 | 0.763 | 0.352 |
| +Ours | **1.000** | **0.976** | **0.349** | **1.000** | **0.987** | **0.358** | **1.000** | **0.969** | **0.353** |

We further provide the proof for the **generalized multi-bit setting** as follows.

*Proof.* Let $\mathbf{B} = (b_1, \ldots, b_m) \in \{0,1\}^m$ be independent, uniform watermark bits, and define

$$k(\mathbf{B}) = \sum_{j=1}^{m} b_j \cdot 2^{m-j} \in \{0, \ldots, 2^m - 1\},$$

We define $k(\mathbf{B})$ as the integer index corresponding to the binary vector $\mathbf{B}$, establishing a one-to-one mapping between $\mathbf{B}$ and its interval $R_{k(\mathbf{B})}$. Partition $k$ into *equiprobable* intervals

$$R_k = \left( \Phi^{-1}\left(\frac{k}{2^m}\right), \ \Phi^{-1}\left(\frac{k+1}{2^m}\right) \right), \qquad k = 0, 1, \ldots, 2^m - 1,$$

so that $\Pr[Z \in R_k] = 2^{-m}$, where $\Phi$ and $\Phi^{-1}$ denote the standard-normal CDF and its inverse.

Define the conditional density of a latent element $Z$ as

$$p_{Z|\mathbf{B}}(z \mid \mathbf{b}) = \begin{cases} 2^m \cdot \phi(z), & \text{if } z \in R_{k(\mathbf{b})}, \\ 0, & \text{otherwise,} \end{cases}$$

where $\phi(z) = \dfrac{1}{\sqrt{2\pi}} e^{-z^2/2}$.

Regardless of $m$, the marginal distribution of $Z$ remains standard normal:

$$Z \sim \mathcal{N}(0,1).$$

By marginalizing over $\mathbf{B}$,

$$\begin{aligned} p_Z(z_T^e) &= \sum_{\mathbf{b} \in \{0,1\}^m} p_{Z|\mathbf{B}}(z_T^e \mid \mathbf{b}) \, P(\mathbf{B} = \mathbf{b}) \\ &= \sum_{k=0}^{2^m-1} 2^m \, \phi(z_T^e) \, \mathbf{1}_{R_k}(z_T^e) \cdot \frac{1}{2^m} \\ &= \phi(z_T^e) \sum_{k=0}^{2^m-1} \mathbf{1}_{R_k}(z_T^e) \\ &= \phi(z_T^e), \end{aligned}$$

since $\{R_k\}$ forms a disjoint cover of $\mathbb{R}$, the sum of indicator functions equals 1. $\qquad\square$

Table 11: Runtime comparison among different watermarking methods.

| Method | Embed | Detect |
| --- | --- | --- |
| Diffusion Model | 1.84s (Diffusion) | 1.25s (Inversion) |
| Tree-Ring | 0.0060s | 0.0008s |
| Gaussian Shading | 1.57s | 0.0021s |
| **Ours** | 0.0280s | 0.0017s |

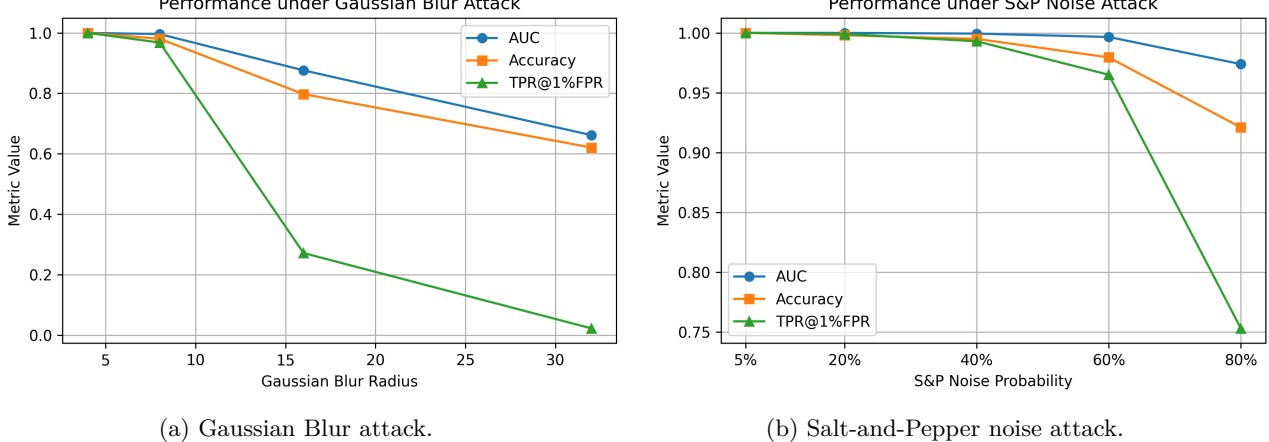

(a) Gaussian Blur attack.                 (b) Salt-and-Pepper noise attack.

Figure 8: Performance degradation under increasing attack strength. **Left:** AUC, Accuracy, and TPR@1%FPR curves under Gaussian Blur with increasing blur radius. **Right:** AUC, Accuracy, and TPR@1%FPR curves under Salt-and-Pepper noise with increasing noise probability.

## B.2 Proof of Proposition 2

*Proof.* We first show that the correlation between any two different elements $z_T^{ijk}$ and $z_T^{i'j'k'}$ in the tensor is given by

$$\text{Corr}(z_T^{ijk}, z_T^{i'j'k'}) = \frac{2}{\pi} \cdot \frac{p-1}{|z_T|-1}.$$

Consider two elements $X$ and $Y$ within a latent representation divided into $p$ patches, each containing $n$ pixels, totaling $|z_T| = pn$ pixels. Pixels within the same patch share a common watermark bit $W$, while those in different patches have independent watermark bits.

$$\text{Cov}(X, Y) = \mathbb{E}[XY] - \mathbb{E}[X]\mathbb{E}[Y].$$

Given $\mathbb{E}[X] = \mathbb{E}[Y] = 0$, this simplifies to:

$$\text{Cov}(X, Y) = \mathbb{E}[XY].$$

When $X$ and $Y$ are in the same position across different patches, they share the same watermark bit $W$. Given $W$, $X$ and $Y$ are independent and follow a Half-Normal Distribution with variance $\sigma^2 = 1$.

The probability density function (PDF) of a Half-Normal Distribution with variance $\sigma^2 = 1$ is

$$f_Y(y) = \sqrt{\frac{2}{\pi}} \exp\left(-\frac{y^2}{2}\right), \quad y \geq 0.$$

The expectation $\mathbb{E}[Y]$ is calculated as

$$\mathbb{E}[Y|W=1] = \int_0^\infty y \cdot f_Y(y)\, dy$$

$$= \sqrt{\frac{2}{\pi}} \int_0^\infty \sqrt{2t} \cdot e^{-t} \cdot \frac{1}{\sqrt{2t}}\, dt \quad \text{(where } t = \frac{y^2}{2}\text{)}$$

$$= \sqrt{\frac{2}{\pi}}.$$

Similarly, for $W = 0$,

$$\mathbb{E}[Y|W=0] = -\sqrt{\frac{2}{\pi}}.$$

Since $X$ and $Y$ are independent given $W$,

$$\mathbb{E}[XY|W] = \mathbb{E}[X|W]\mathbb{E}[Y|W] = \left(\sqrt{\frac{2}{\pi}}\right)^2 = \frac{2}{\pi}.$$

When $X$ and $Y$ are in different positions across different patches, their watermark bits $W_X$ and $W_Y$ are independent. Therefore,

$$\mathbb{E}[XY|\text{Different Positions}] = \mathbb{E}[X]\mathbb{E}[Y] = 0.$$

The probability that two randomly selected pixels are in the same position across different patches is

$$P_s = \frac{\#\text{Same Position}}{\#\text{Total}} = \frac{p-1}{np-1}$$

Combining the cases, we have

$$\mathbb{E}[XY] = P_s \cdot \frac{2}{\pi} + (1 - P_s) \cdot 0 = \frac{2}{\pi} \cdot \frac{p-1}{np-1}.$$

Given that $\text{Var}(X) = \text{Var}(Y) = 1$, the correlation $\rho$ is:

$$\rho = \text{Corr}(X, Y) = \frac{\text{Cov}(X,Y)}{\sqrt{\text{Var}(X)\text{Var}(Y)}} = \frac{2}{\pi} \cdot \frac{p-1}{np-1}.$$

$$\square$$

Furthermore, we prove the normalized Bures–Wasserstein (BW) distance $\hat{d}_{\text{BW}}$ between the watermarked sampling distribution and $\mathcal{N}(0, I)$ is at most

$$\hat{d}_{\text{BW}}(z_T, \mathcal{N}(0, I)) \leq \rho = \frac{2}{\pi} \cdot \frac{p-1}{|z_T| - 1}.$$

Let $\Sigma$ be the covariance matrix of a centered Gaussian latent distribution with correlation parameter $\rho$ as defined in our paper. Then the normalized Bures–Wasserstein distance $\hat{d}_{\text{BW}}$ between this distribution and the standard normal $\mathcal{N}(0, I_{np})$ satisfies

$$\hat{d}_{\text{BW}} = \frac{BW^2}{np} < \rho,$$

i.e., the normalized Bures–Wasserstein distance ($BW^2$) is upper bounded by the correlation $\rho$.

$$BW^2 = \text{Tr}(\Sigma) + \text{Tr}(I_{np}) - 2\text{Tr}(\Sigma^{1/2})$$

is the squared Bures–Wasserstein distance between a Gaussian distribution with covariance $\Sigma$ and the standard normal $\mathcal{N}(0, I_{np})$. Consider a spiked covariance structure

$$\Sigma = (1 - \rho)I_{np} + \rho \mathbf{1}\mathbf{1}^\top, \qquad \mathbf{1} = (1, \ldots, 1)^\top \in \mathbb{R}^{np},$$

so that $\mathrm{Tr}(\Sigma) = np$, which is a constant denotes the size of $z_T$.

Factor out $(1 - \rho)$:

$$\Sigma = (1 - \rho)\big(I_{np} + uu^\top\big), \qquad u = \sqrt{\frac{\rho}{1 - \rho}}\mathbf{1},$$

where $u^\top u = \frac{\rho}{1-\rho}np = \Theta(np)$.

For a rank-1 perturbation $S = I + uu^\top$, a rank-1 update admits a closed-form square root:

$$S^{1/2} = I + \frac{\sqrt{1 + u^\top u} - 1}{u^\top u}uu^\top.$$

Hence

$$\Sigma^{1/2} = \sqrt{1 - \rho}I_{np} + \mathcal{O}((np)^{-1})\mathbf{1}\mathbf{1}^\top,$$

and its trace is

$$\mathrm{Tr}(\Sigma^{1/2}) = \sqrt{1 - \rho}np + \mathcal{O}(1).$$

Plugging this into the BW formula gives

$$BW^2 = np + np - 2\big(\sqrt{1 - \rho}np + \mathcal{O}(1)\big) = 2np(1 - \sqrt{1 - \rho}) - \mathcal{O}(1).$$

Using the identity

$$1 - \sqrt{1 - \rho} = \frac{\rho}{1 + \sqrt{1 - \rho}},$$

we obtain

$$BW^2 = 2np\frac{\rho}{1 + \sqrt{1 - \rho}} - \mathcal{O}(1) < np\rho.$$

Thus, for large $np$,

$$\boxed{\frac{BW^2}{np} < \rho},$$

meaning the normalized Bures–Wasserstein distance is bounded by $\rho$.

### B.3 Illustration of the Statistical Test

In this work, our primary focus is on evaluating the actual performance of the watermarking method. However, a statistical analysis can also be derived. Let $m \in \{0,1\}^k$ represent a $k$-bit (independent) watermark embedded in the model. We extract the message $m'$ from an image $x$ and compare it with $m$. As outlined in previous works, the detection test is based on the number of matching bits, $A(m, m')$. Specifically, if

$$A(m, m') \geq \tau \quad \text{where} \quad \tau \in \{0, \ldots, k\},$$

then the image is flagged. This approach provides a level of robustness against imperfections in the watermarking process.

Formally, we test the statistical hypothesis $H_1$: "image $x$ was generated by the watermarked model" against the null hypothesis $H_0$: "image $x$ was not generated by the watermarked model." Under $H_0$ (i.e., for non-watermarked images), we assume that the bits $m'_1, \ldots, m'_k$ are independent and identically distributed (i.i.d.) Bernoulli random variables with a parameter of 0.5. Consequently, $A(m, m')$ follows a binomial distribution with parameters $(k, 0.5)$. This assumption has been experimentally validated.

The theoretical FPR is defined as the probability that $A(m, m')$ exceeds the threshold $\tau$. It is calculated using the CDF of the binomial distribution. A closed-form expression can be derived using the regularized incomplete beta function $I_x(\alpha; \beta)$:

$$\text{FPR}(\tau) = \mathbb{P}(M > \tau | H_0) = I_{1/2}(\tau + 1, k - \tau).$$

## C More on Visualization

### C.1 More Generation Results

In this section, we provide additional generation results to illustrate that our watermarking method maintains comparable generation quality. Results for Stable-Diffusion prompts are presented in Table 12 and Table 13, while generation results for MS-COCO captions, including the textual prompts, generated images, and corresponding realistic images from the MS-COCO dataset, are shown in Table 14 and Table 15.

All generated images effectively capture the information conveyed by the textual prompts. In some cases, the watermarked images show slight improvements in quality (e.g., the 1st prompt from Stable-Diffusion), while others exhibit minor degradation (e.g., the 2nd prompt from MS-COCO), which can be attributed to the inherent randomness in the input Gaussian representation. These examples illustrate the robustness of our watermarking approach, as the visual quality of the generated images remains consistent, and the watermarking process does not interfere with the generation fidelity.

### C.2 Failed Examples

We report in Table 16 the cases with the largest CLIP-Score reduction when compared to their non-watermarked counterparts. These examples represent the worst deviations observed in our evaluation and were selected specifically because they exhibit the largest semantic drift under watermark injection. They primarily occur under prompts that involve extreme lighting, heavy render-like effects, or unusually complex artistic constraints. These conditions make both inversion and generation more unstable, which amplifies small perturbations introduced by watermarking. Even in these cases, the semantic content of the prompt is generally preserved, but fine-grained structures or lighting consistency may deviate. Understanding and mitigating these extreme cases can be an interesting direction for future work.

## D    Broader Impact Statement and Limitations

**Broader Impact Statement.**    Our work aims to enhance the safety and accountability of diffusion-based generative models by enabling reliable watermarking. This helps detect AI-generated content, prevent misuse, and promote responsible deployment in real-world applications. DiffKGW does not encode any user-specific information, ensuring that it cannot facilitate user-level tracking. While the approach improves robustness in both text-to-image and image-to-image settings, we also acknowledge that image-to-image editing may introduce ambiguity. We have highlighted this limitation and believe it represents an interesting direction for future research.

**Limitations.**    Our method primarily focuses on robustness against spatial perturbations (e.g., cropping, rotation) and common post-processing operations. However, it may be less effective under extreme distribution shifts or adversarial removal attacks specifically crafted to disrupt the frequency-domain signal used in watermarking. Additionally, our detection framework assumes access to a partial or full reverse diffusion process, which may not always be feasible in real-world scenarios with closed-source models.

## E    GenAI Usage Disclosure

During the preparation of this manuscript, we made controlled use of large language models (LLMs), specifically ChatGPT, as an auxiliary writing tool. The LLM was used solely for stylistic refinement, such as improving the fluency, grammar, and readability of paragraphs originally drafted by the authors. All scientific content, including the conceptual development, methodology, experimental design, and main narrative of the paper, was fully conceived, written, and validated by the authors without reliance on LLMs. Therefore, the LLMs served exclusively in a supportive editing role.

| Prompt | Non-watermarked | Watermarked |
|---|---|---|
| a portrait of a girl skull face, marilyn monroe, in the style of artgerm, charlie bowater, atey ghailan and mike mignola, vibrant colors and hard shadows and strong rim light, plain background, comic cover art, trending on artstation |  |  |
| a very beautiful anime cute girl, full body, long wavy blond hair, sky blue eyes, full round face, short smile, fancy top, miniskirt, front view, medium shot, mid-shot, laying in bed, highly detailed, cinematic wallpaper by Stanley Artgerm Lau |  |  |
| overgrown foliage overtaking massive japanese temples, underwater environment, borealis, scenery, professional, award-winning, trending on artstation, hyper detailed, realistic, beautiful, emotional, shiny, golden, picture |  |  |
| a high detail photograph of manhattan after being destroyed by an alien race, building, avenue, urban architecture, americana architecture, concrete architecture, paved roads, by thomas kinkade trending on artstation, photorealistic, wild vegetation, utopian, futuristic, blade runner |  |  |
| digital detailed portrait of anthromorphic female hyena, in style of zootopia, fursona, furry, furaffinity, 4 k, deviantart, wearing astronaut outfit, in style of disney zootopia, floating in space, space background, in deep space, dark background, hyena fursona, cyberpunk, female, detailed face, style of artgerm |  |  |

Table 12: Stable-Diffusion Prompts with corresponding non-watermarked and watermarked images. All images well capture the textual prompt information. Some watermarked images demonstrate slight improvements (e.g., the 5th) compared to the non-watermarked versions, while some show minor degradation (e.g., the 4th), attributed to the randomness inherent in the input Gaussian representation.

| Prompt | Non-watermarked | Watermarked |
|---|---|---|
| movie poster, game about deep caves and void monsters, cinematic light, clean linework, finely detailed, 4 k, trending on artstation, concept art by stanley lau |  |  |
| a simple micro-service deployed to a public cloud, security, attack vector, trending on Artstation, painting by Jules Julien, Leslie David and Lisa Frank, muted colors with minimalism |  |  |
| symmetry!! portrait of a female sorcerer, dark fantasy, intricate, elegant, highly detailed, my rendition, digital painting, artstation, concept art, smooth, sharp focus, illustration, art by artgerm and greg rutkowski and alphonse mucha and huang guangjian and gil elvgren and sachin teng |  |  |
| pastel landscape of an anime field. clean sharp digital art, environment concept art, by rossdraws, ghibli, breath of the wild, greg rutkowski |  |  |
| a tiny worlds by greg rutkowski, sung choi, mitchell mohrhauser, maciej kuciara, johnson ting, maxim verehin, peter konig, bloodborne, 8 k photorealistic, cinematic lighting, hd, high details, dramatic, dark atmosphere, trending on artstation |  |  |

Table 13: Stable-Diffusion Prompts with corresponding non-watermarked and watermarked images. All images well capture the textual prompt information. Some watermarked images demonstrate slight improvements (e.g., the 2nd) compared to the non-watermarked versions, while some show minor degradation (e.g., the 3rd), attributed to the randomness inherent in the input Gaussian representation.

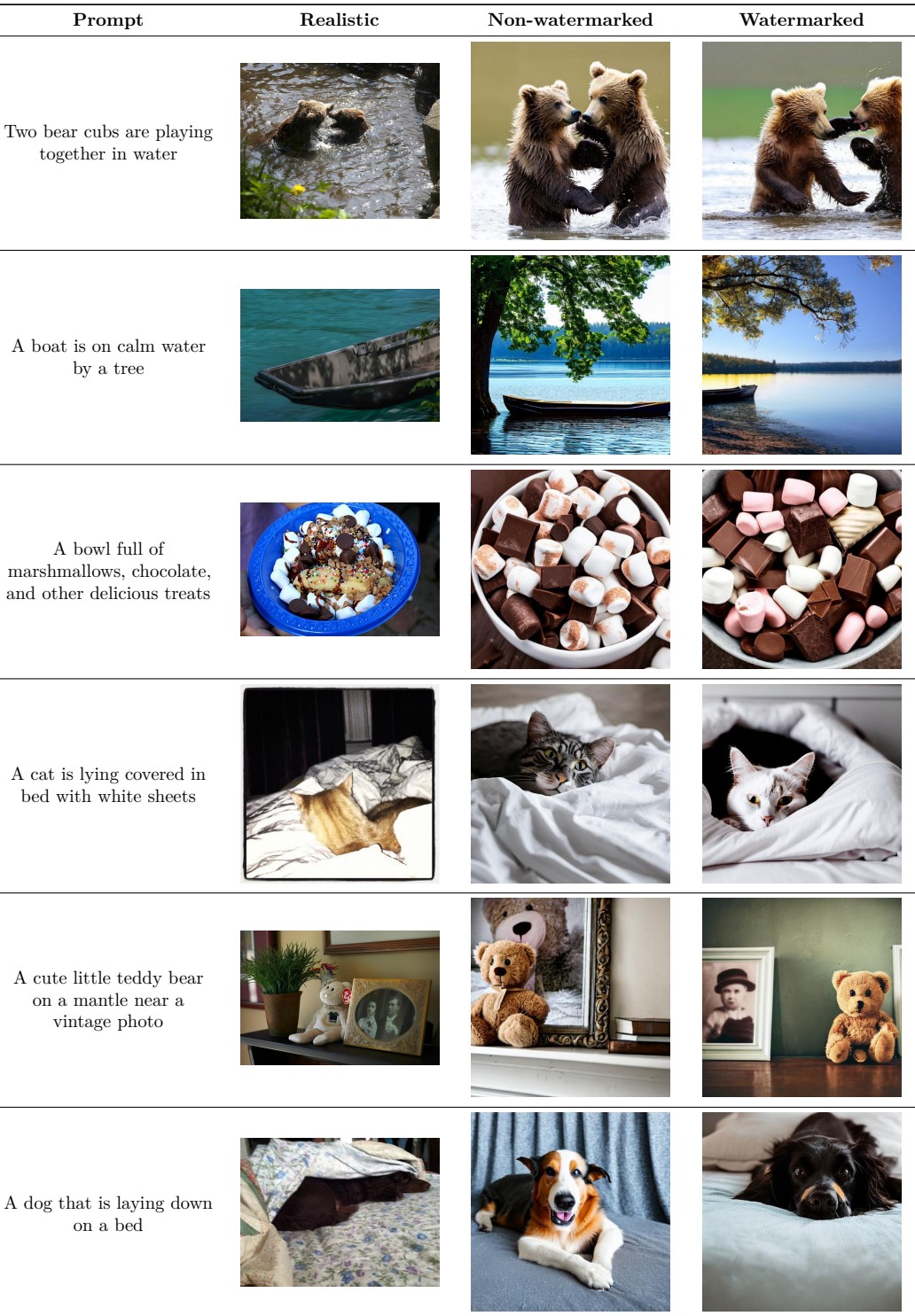

Table 14: MS-COCO Captions with corresponding realistic, non-watermarked, and watermarked images. All generated images capture the textual prompt information well. Some watermarked images demonstrate slight improvements (e.g., the 4-6th) compared to the non-watermarked versions, while some show minor degradation (e.g., the 1st), attributed to the randomness inherent in the input Gaussian representation.

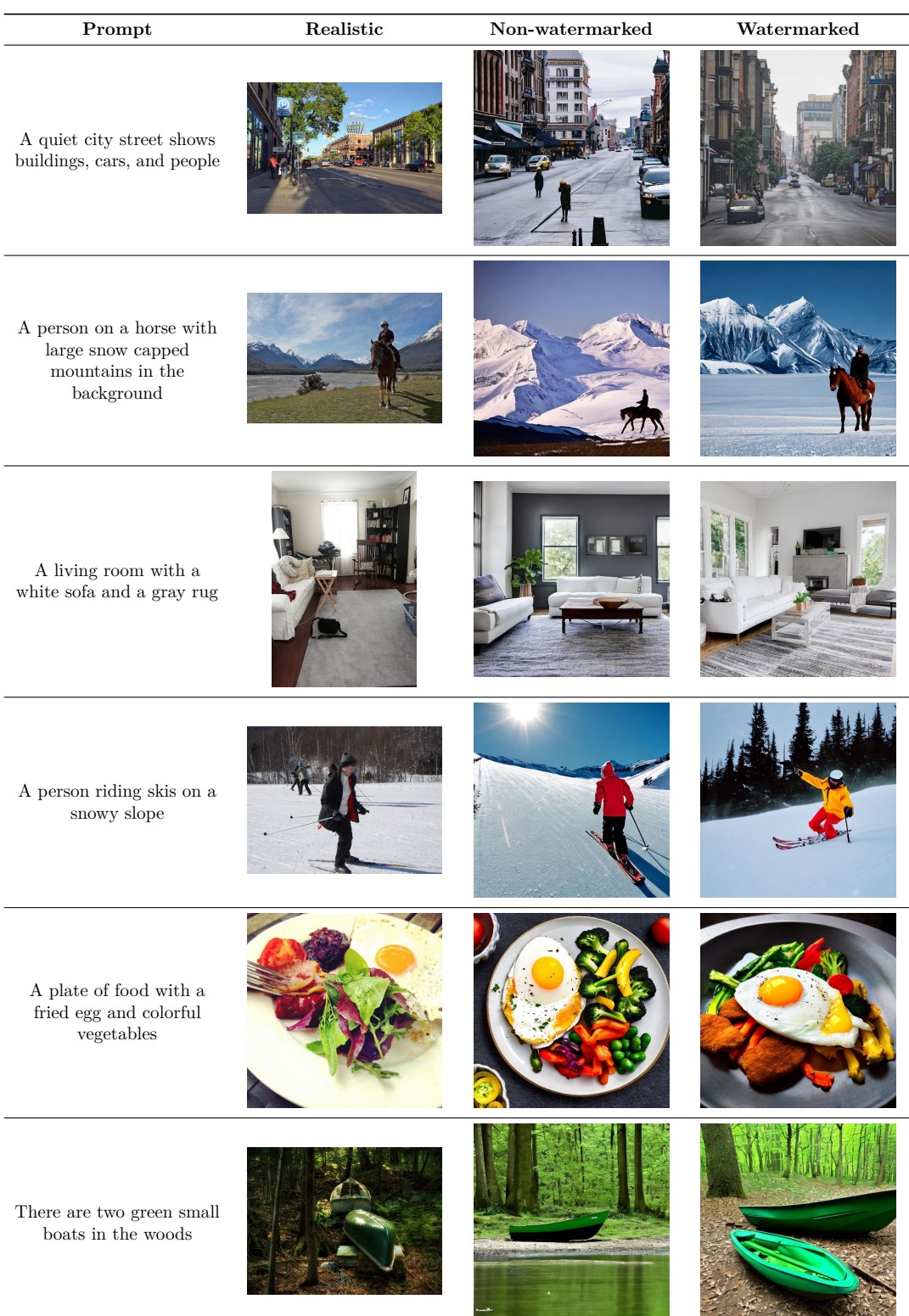

Table 15: MS-COCO Captions with corresponding realistic, non-watermarked, and watermarked images. All generated images capture the textual prompt information well. Some watermarked images demonstrate slight improvements (e.g., the 1st) compared to the non-watermarked versions, while some show minor degradation (e.g., the 2nd), attributed to the randomness inherent in the input Gaussian representation.

| Prompt | Non-watermarked | Watermarked |
|---|---|---|
| profile portrait, helmet tiger cyberpunk made of pink lava and fire design by mark brooks and brad kunkle detailed, aurora digital package, profile portrait, cyberpunk fashion, realistic shaded perfect face, fine details, very dark environment, misty atmosphere, closeup, d & d, fantasy, intricate, elegant, highly detailed, digital painting, artstation, concept art, matte, sharp focus, illustration, hearthstone | | |
| a lighthouse in space, meteors, air shot, elegant, digital painting, concept art, smooth, sharp focus, illustration, from StarCraft by Ruan Jia and Mandy Jurgens and Artgerm and William-Adolphe Bouguerea | | |
| portrait art of nicole aniston 8 k ultra realistic, lens flare, atmosphere, glow, detailed, intricate, full of colour, cinematic lighting, trending on artstation, 4 k, hyperrealistic, focused, extreme details, unreal engine 5, cinematic, masterpiece | | |

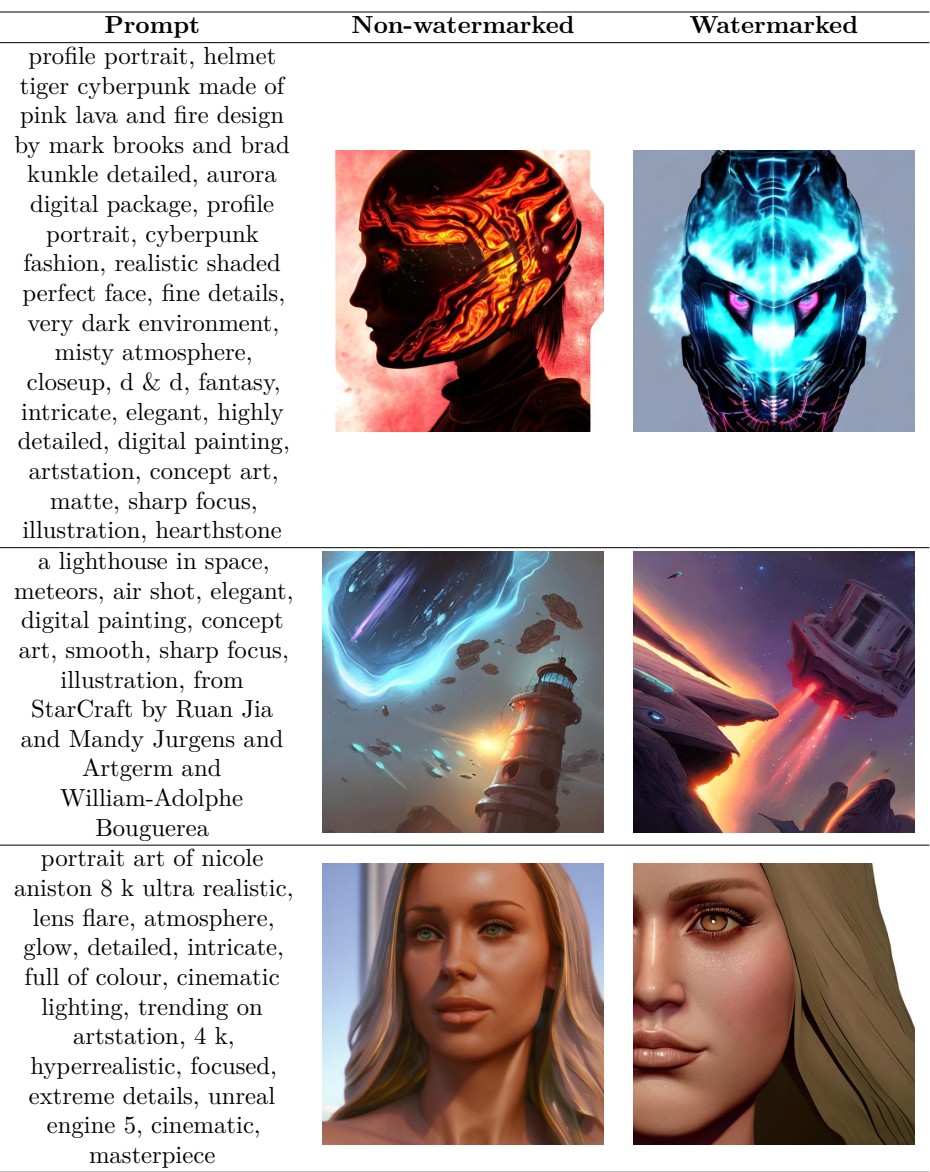

Table 16: Failure cases under highly stylized prompts. These cases typically involve extreme lighting, cinematic rendering, or heavy stylistic constraints, where both inversion and generation become unstable.

