# OpenReview forum: "DiffKGW: Stealthy and Robust Diffusion Model Watermarking"
_TMLR — Accepted by TMLR_

### Review · Reviewer_5zGy · 2025-11-04

**Summary Of Contributions:**

This paper presents DiffKGW, a novel, training-free watermarking method for diffusion models. The core contributions are: (1) a "unified framework" that categorizes latent watermarking methods; (2) the generalization of the KGW watermark from LLMs to the continuous latent space of diffusion models, which aims to preserve the initial Gaussian distribution; and (3) a hybrid shape design combining "Gaussian Ring" and "Random Gaussian" methods to provide robustness against both geometric and noise-based attacks.

**Audience:**

Yes

**Audience Explanation:**

The paper tackles provenance, copyright, and misuse of diffusion models. This is one of the most critical and high-priority "AI safety" and "AI ethics" problems currently facing the machine learning community. It directly addresses diffusion models, which remain a cutting-edge and intensely studied area of generative AI. The core idea of adapting a watermark from LLMs to the continuous latent space of diffusion models is a novel and clever contribution.

**Broader Impact Concerns:**

1. It may enable user-level tracking without users' consent
2. False attribution and proof of ownership due to the image-to-image watermark.

**Claims And Evidence:**

Yes

**Claims Explanation:**

The paper's primary claim is that DiffKGW is "stealthy" because it "preserves the Gaussian distribution of [the] initial latent representation." The paper provides high fidelity scores (FID, CLIP-Score) to show the output image quality is high. It also provides a theoretical analysis (Lemma 1, Proposition 2).

The hybrid "Gaussian Ring" and "Random Gaussian" method is robust to a comprehensive set of attacks, including geometric (rotation) and noise-based (JPEG, blur) attacks. Table 1 shows high detection accuracy against one fixed-parameter version of each attack (e.g., "rotation by 75 degrees," "JPEG compression with a ratio of 25%").

The paper introduces a "unified framework" for watermarking as a key contribution. Section 3 categorizes existing watermarking methods into three dimensions (element sampling, shape design, channel selection).

**Requested Changes:**

1. The current evidence for stealthiness (FID, CLIP-Score) measures image fidelity. The author should clarify the stealthiness concept in watermark, is it regarding the perceptual quality, data distribution with respect to the generated image distribution or natural image distribution? Though the author defined the watermark distribution in Proposition 2, the unit Gaussian distribution lacks clarity.

2. The author proposed two ways of watermarking in different channels of the images and stated them as redundancy design. However, the empirical results didn’t show the robustness of each individual watermarking method. Can the author provide some empirical analysis of the effectiveness of the redundancy design, more specifically, is one watermark (ring) always better than the other (random)?

3. The selection of the optimal watermark channel seems to be arbitrary. Can the author provide the algorithm and justification for the channel selection, some empirical analysis would be sufficient.

4. Section 3 uses a unified framework in the section title may be over claim. A taxonomy or a structured literature review may be more appropriate.

---

> ### Author Response · Authors · 2025-12-12
>
> Dear Reviewer 5zGy,
>
> We sincerely thank the reviewer for recognizing the contributions of our work and for providing thoughtful and constructive feedback. Your comments helped us clarify the definition of stealthiness, strengthen the empirical justification, and refine the presentation of our taxonomy. All corresponding revisions have been incorporated into the updated manuscript and are highlighted in blue for clarity. We address each request below.
>
> ## **Requested Change 1**
>
> > The current evidence for stealthiness (FID, CLIP-Score) measures image fidelity. The author should clarify the stealthiness concept… the unit Gaussian distribution lacks clarity.
>
> **Response:**
>
> We thank the reviewer for pointing out the need for a clearer definition of stealthiness. In our context, stealthiness means that the watermark should be visually imperceptible and that watermarked and non-watermarked images should remain indistinguishable in distribution. Achieving this requires preserving the sampling distribution of the latents during watermark injection. We have revised Proposition 2 to clarify this point, the “unit Gaussian distribution” refers to the standard normal latent distribution used by the original, unwatermarked diffusion model.
>
> ---
>
> ## **Requested Change 2**
>
> > The author proposed two ways of watermarking… empirical results didn’t show the robustness of each… Can the author provide some empirical analysis?
>
> **Response:**
> We thank the reviewer for raising this point. We have already included empirical analyses in Appendix A.5 and Table 8 to evaluate the robustness contributed by each of the two watermark components. The results show that both the Random Gaussian and Gaussian Ring patterns are essential to the overall performance of our method. The Random Gaussian component provides stronger robustness against noise-based attacks, as evidenced by high detection performance under noise settings. In contrast, the Gaussian Ring component exhibits greater resilience to geometric transformations such as rotation. These complementary strengths justify the use of both components in our hybrid design.
>
> ---
>
> ## **Requested Change 3**
>
> > The selection of the optimal watermark channel seems to be arbitrary… provide the algorithm and justification.
>
> **Response:**
> We thank the reviewer for raising this concern. The full procedure for selecting the optimal watermark channel is already provided in Algorithm 3 in the Appendix. Our method first evaluates channels and then injects the watermark into the selected one. As shown in Figure 6 (Left) in the submitted main paper, there is substantial variation in performance across channels, with Channel 2 achieving the highest accuracy. This empirical observation motivates our design choice and provides justification for the selected channel.
>
> ---
>
> ## **Requested Change 4**
>
> > Section 3 uses a unified framework… may be over claim. A taxonomy or structured review may be more appropriate.
>
> **Response:**
> We thank the reviewer for the helpful suggestion. We have revised the terminology in Section 3 to “taxonomy” to accurately reflect our intent of structuring and organizing prior work without overstating the contribution. We appreciate this clarification and believe it improves the precision of our presentation.
>
> ---
>
>
> ## **Broader Impact Concerns**
>
> > It may enable user-level tracking without users' consent.
>
> **Response:**
> We thank the reviewer for raising these important broader-impact considerations. DiffKGW is mainly designed as a model-level identification watermark: the watermarking mechanism is fixed at model generation time, cannot be modified or personalized by users, and does not encode any user-specific information. Consequently, it cannot be used for user-level tracking without consent. We have added explicit clarification of this point in the Broader Impact section in Appendix D.
>
> > False attribution and proof of ownership due to the image-to-image watermark.
>
> **Response:**
> The concern regarding false attribution and ownership verification in the image-to-image setting is indeed insightful. Our main focus is to improve robustness in both text-to-image and image-to-image scenarios. However, because image-to-image methods alter the generated content, reliably using such watermarks remains challenging and may introduce ambiguity. This limitation also highlights an interesting direction for future research based on our paper. We have added a clarification of this point in the Broader Impact section in Appendix D.
>
> ---
>
> We appreciate the reviewer’s insightful comments and believe the revisions have further improved the clarity and strength of the paper. Should you have any further questions, we are more than happy to discuss.

---

> > ### Comment · Reviewer_5zGy · 2025-12-16
> >
> > Thanks for authors detailed responses. I am sorry for the missing details in your appendix. I have no further questions.

---

> > > ### Author Response · Authors · 2025-12-16
> > >
> > > We thank the reviewer for the clarification and understanding. We are glad that our responses were helpful in addressing the concerns. We sincerely appreciate the reviewer’s time and effort, which helped improve the quality of this work.

---

### Review · Reviewer_9EFc · 2025-11-19

**Summary Of Contributions:**

The work introduces a generalisation of the LLM KGW watermarking scheme to diffusion models.

Strengths:
- mostly clear presentation
- elegant generalisation
- performant scheme that maintains high fidelity
- good evaluation of basic robustness

Weaknesses:
- unclear threat model
- robustness evaluation should show how/when the scheme breaks, some attacks details are missing
- attack evaluation beyond image editing (watermark destruction) missing

**Additional Comments:**

Evaluating diffusion-based attack is a good first towards testing the robustness of the scheme. Additional things that you can consider:
- recovery of the ring from degenerate samples (e.g. uni colors) or from minimally edited images (e.g. in model based online image editing)
- reuse of watermarks
- robustness under watermark overwriting

**Audience:**

Yes

**Audience Explanation:**

Identification of generative content as well as neural-based watermarking are both topics of interest to the broader machine learning and media communities. This work generalises an existing scheme for watermarking LLM generated text to image generation.

**Claims And Evidence:**

Yes

**Claims Explanation:**

The work provides a sufficient evidence to back up the claims of the effectiveness of the scheme in benign conditions; as well as its robustness to weak transformations (e.g. noising, compression), and one strong attacks (diffusion-based regeneration) in a limited setting.

**Requested Changes:**

1. Watermarking as a whole is a broad field with watermarks that serve different purpose. Based on the selection of related work I can tell that the authors intend to __identify__ watermarked content. However, this should be made more explicit to avoid confusion with brittle, or attribution watermarking. In other words, you'd make the threat model more explicit.
2. Some presentation of worst case (degenerate) examples would help the readers understand the failure modes more.
3. It isn't clear to me how strong the diffusion-based regeneration is either -- the appendix doesn't provide details beyond citing the work. More broadly, proper robustness evaluation needs to show when a given method breaks the scheme not just that it doesn't in some arbitrary setting. E.g. blurring or noising is guaranteed to destroy the watemark if it's sufficiently strong (at the cost of image quality).
4. As the authors noted in the evaluation, the adversarial evaluation is quite limited. I can accept that if more explicit discussion around potential attacks is provided in the main body -- the limitations discussed in the appendix are easy to miss and too abstract. See additional comments.

---

> ### Author Response · Authors · 2025-12-12
>
> Dear Reviewer 9EFc,
>
> We sincerely thank the reviewer for recognizing the contributions of our work and for providing insightful and constructive feedback. Your comments have helped us improve both the clarity of the presentation and the robustness analysis, and we have incorporated all corresponding revisions into the updated manuscript. All modifications are highlighted in blue for clarity. Below, we respond to each point in detail.
>
>
> ---
>
> ## **Weakness 1 \& Requested Change 1**
>
> > Watermarking as a whole is a broad field with watermarks that serve different purpose. Based on the selection of related work I can tell that the authors intend to identify watermarked content. However, this should be made more explicit to avoid confusion with brittle, or attribution watermarking. In other words, you'd make the threat model more explicit.
>
>
> **Response:**
> We thank the reviewer for the helpful suggestion. We have revised the manuscript to explicitly clarify that the goal of our watermarking scheme is watermark identification, rather than brittle or attribution watermarking. The threat model has been made more explicit in the introduction and methodology sections to avoid ambiguity. These clarifications are now included in the updated version of the paper.
>
> ---
>
> ## **Requested Change 2**
>
> > Some presentation of worst case (degenerate) examples would help the readers understand the failure modes more.
>
> **Response:**
> We have added degenerate examples in a new subsection (Appendix C.2) and Table 16. These cases correspond to the largest CLIP-Score reductions relative to their non-watermarked counterparts, representing the worst deviations observed in our evaluation. They primarily arise from prompts involving extreme lighting, heavy render-like effects, or unusually complex artistic constraints. Such settings make both inversion and generation inherently unstable, which in turn amplifies small perturbations introduced by watermarking. Even in these cases, the core semantic content of the prompt is largely preserved, though fine-grained structures or lighting details may deviate. Understanding and mitigating these extreme failure cases presents a valuable direction for future work.
>
> ---
>
> ## **Weakness 2 \& Requested Change 3**
>
> > It isn't clear to me how strong the diffusion-based regeneration is… proper robustness evaluation needs to show when a given method breaks the scheme.
>
> **Response:**
> We thank the reviewer for the insightful comment. We have expanded Appendix A.1 to include the full protocol of the diffusion-based regeneration process for clarity. In addition, we added Appendix A.9 and Figure 8 to explicitly analyze failure conditions by progressively increasing the intensity of blur and noise. As expected, stronger corruptions eventually degrade the watermark signal because they severely distort the underlying image. Nonetheless, even under extremely destructive settings, such as replacing 80% of the pixels with noise, our method still maintains a reasonably detection accuracy. These results illustrate both where the watermark begins to break and how resilient it remains under heavy degradation.
>
> ---
>
> ## **Weakness 3 \& Requested Change 4**
>
> > As the authors noted… adversarial evaluation is quite limited… more explicit discussion around potential attacks should be provided in the main body.
>
> **Response:**
>
> We thank the reviewer for highlighting the importance of broader adversarial considerations. We have expanded the discussion in the experimental part of the main body to more explicitly cover potential attack vectors and limitations, including several directions suggested by the reviewer. In particular, we now discuss recovery of the ring structure from degenerate or minimally edited images, potential risks related to watermark reuse, and robustness under watermark overwriting. These aspects represent meaningful avenues for future investigation and help situate our method within a more complete adversarial landscape.
>
> ---
>
> We appreciate the reviewer’s insightful comments and believe the revisions have further improved the clarity and strength of the paper. Should you have any further questions, we are more than happy to discuss.

---

> > ### Comment · Reviewer_9EFc · 2025-12-14
> >
> > Thank you for the response, and I'm satisfied with these changes. Minor caveat, Appendix A.1 should still explain the parametrisation of the diffusion process -- "applies several latent diffusion steps" is too vague.

---

> > > ### Author Response · Authors · 2025-12-14
> > >
> > > Thank you for your response, and we are glad to hear that you are satisfied with the revisions. We apologize for missing this detail in the previous version. In Appendix A.1, the diffusion process indeed applies fifty latent diffusion steps, consistent with the generation procedure used throughout our experiments. We have revised the appendix to explicitly state this parametrization and clarify the diffusion setup.

---

### Review · Reviewer_wv9s · 2025-11-27

**Summary Of Contributions:**

The paper investigates watermarking for diffusion models. Inspired by the KGW watermarking method used in Large Language Models (LLMs), the authors propose DiffKGW, a method that partitions the standard Gaussian distribution into "Green" and "Red" regions based on a secret key. The latent representations are then sampled exclusively from the Green regions. The authors theoretically show that the method preserves the Gaussian distribution of the representations.

Furthermore, the paper introduces a "unified framework" categorizing watermarks into three dimensions: element sampling, shape design, and channel selection. Based on this framework, they implement a hybrid strategy that dynamically selects between "Gaussian Rings" and "Random Gaussian" patterns based on the sensitivity to geometric transformations. Experiments on Stable Diffusion and InstructPix2Pix demonstrate that the method achieves high visual quality and robustness against both geometric and noise-based attacks

Strengths:
- The adaptation of the discrete KGW in LLM to the continuous latent space of diffusion models is mathematically sound. The result of Lemma 1 provides a theoretical contribution that distinguishes this work from heuristic methods like Tree-Ring [1] which might distort the latnet sapce.
- The categorization of watermarking methods into element sampling, shape design, and channel selection provides a useful taxonomy for understanding the design space of diffusion watermarking.
- The method demonstrates strong performance across a diverse set of attacks and maintains high visual quality.

Weakness:
- The paper has limited novelty. While the distribution-preserving sampling is well-motivated, the core mechanism is a direct translation of the existing LLM KGW method into the diffusion domain. Additionally, the "Gaussian Ring" shape design bears significant similarity to prior works such as Tree-Ring and Ring-ID.
- The "Gaussian Random" watermarking component is novel, but the ablation studies suggest limited practical impact. Table 8 shows that adding the Random Gaussian component improves the average detection rate from 0.952 (using only Gaussian Rings) to 0.984. The authors should better justify whether the complexity of the hybrid strategy is necessary, given these diminishing returns.

[1] Yuxin Wen, John Kirchenbauer, Jonas Geiping, and Tom Goldstein. Tree-rings watermarks: Invisible fingerprints for
diffusion images. Advances in Neural Information Processing Systems, 36, 2023.

[2] Hai Ci, Pei Yang, Yiren Song, and Mike Zheng Shou. Ringid: Rethinking tree-ring watermarking for enhanced
multi-key identification. arXiv preprint arXiv:2404.14055, 2024.

**Audience:**

Yes

**Audience Explanation:**

The topic of watermarking diffusion models is critical for protecting copyright and identifying the source of inappropriate images. The proposed method offers a valuable direction for the community by successfully adapting LLM watermarking techniques to diffusion models. Furthermore, the paper provides a unified framework with three clear design principles that can effectively inspire future research.

**Claims And Evidence:**

Yes

**Claims Explanation:**

As stated in the strengths, the proposed method demonstrates an effective balance of robustness and utility. The authors support their claims with accurate theoretical and empirical evidence.

**Requested Changes:**

1. The ablation study in Table 8 indicates that the "Random Gaussian" component provides only a marginal improvement in average detection performance compared to using Gaussian Rings alone. Given the added complexity of the hybrid channel selection strategy, please provide a stronger justification for including this component.

2. In the text-to-image experiments (Table 1), the method is compared against seven baselines. However, the image-to-image section (Table 6) only compares against Tree-Ring and Gaussian Shading. Please extend the image-to-image comparison to include other relevant baselines from the T2I section (such as RivaGan and Stable Signature) to ensure consistency, or explain why they were excluded.

3. Please provide an analysis of the running time (or latency) for both the embedding and extraction steps compared to key baselines. This is important to verify the practical efficiency of the proposed method.

---

> ### Author Response · Authors · 2025-12-12
>
> Dear Reviewer wv9s,
>
> We sincerely thank the reviewer for the thoughtful and constructive feedback, as well as for recognizing the contributions of our work. We have carefully addressed each of the identified weaknesses and incorporated all requested changes into the revised manuscript. All modifications are highlighted in blue for clarity. Below, we provide detailed responses to each point.
>
> ## **Weakness 1**
>
> > While the distribution-preserving sampling is well-motivated, the core mechanism is a direct translation of the existing LLM KGW method into the diffusion domain. Additionally, the "Gaussian Ring" shape design bears significant similarity to prior works such as Tree-Ring and Ring-ID.
>
> **Response:**
>
> We clarify that adapting KGW to diffusion models is not a direct translation. Unlike discrete logit perturbations in LLMs, diffusion watermarking must operate in a continuous Gaussian space and therefore requires a probability-preserving partitioning strategy rather than token-level manipulation. In addition, we address vision-specific challenges through patch-, shape-, and channel-level designs that improve both robustness and stealthiness. Proposition 1 and Proposition 2 formalize the properties in terms of robustness and distribution-preserving stealthiness.
>
> Regarding Tree-Ring and Ring-ID, although all methods employ a ring-shaped region, the underlying mechanisms are fundamentally different. Prior methods enforce constant values within the ring, whereas our approach performs probability-preserving sampling over that region. This distinction leads to substantially improved robustness, particularly under noise and blur attacks, as shown in our experiments.
>
> ---
>
> ## **Weakness 2 \& Requested Change 1**
>
> > The "Gaussian Random" watermarking component is novel, but the ablation studies suggest limited practical impact. Table 8 shows that adding the Random Gaussian component improves the average detection rate from 0.952 (using only Gaussian Rings) to 0.984. The authors should better justify whether the complexity of the hybrid strategy is necessary, given these diminishing returns.
>
> **Response:**
> We thank the reviewer for the positive assessment. While the average improvement in Table 8 may appear modest (0.952 to 0.984) due to the already high performance, the per-attack analyses in Appendix A.8 show that the Gaussian Random component is essential under noise- and blur-based transformations. For instance, the detection rate improves from 0.937 to 1.000 under noise and from 0.923 to 0.996 under blur, substantially increasing reliability in these practically important scenarios. In addition, the watermark sampling process can be precomputed and is highly efficient, as further discussed in our response to Requested Change 3.
>
>
> ---
>
> ## **Requested Change 2**
>
> > In the text-to-image experiments (Table 1), the method is compared against seven baselines. However, the image-to-image section (Table 6) only compares against Tree-Ring and Gaussian Shading. Please extend the image-to-image comparison to include other relevant baselines from the T2I section (such as RivaGan and Stable Signature) to ensure consistency, or explain why they were excluded.
>
> **Response:**
>
> We thank the reviewer for the helpful suggestion. In the paper, we initially included only the two most representative and strongest baselines (Tree-Ring and Gaussian Shading) for the image-to-image setting. Following the reviewer’s request, we have now extended the comparison to include the remaining baselines used in the text-to-image experiments, such as RivaGan and Stable Signature. The updated results are provided in Appendix A.8 and Table 6, and our method continues to outperform all baselines with consistent improvements.
>
> ---
>
> ## **Requested Change 3**
>
> > Please provide an analysis of the running time (or latency) for both the embedding and extraction steps compared to key baselines. This is important to verify the practical efficiency of the proposed method.
>
> **Response:**
> We appreciate the reviewer’s suggestion and have added a detailed runtime comparison for embedding and detection. As shown in Appendix A.8 and Table 11, the computational overhead introduced by our watermark is minimal. Tree-Ring achieves the fastest runtime, but our method also remains highly efficient, requiring only 0.028 s for embedding and 0.0017 s for detection. In contrast, diffusion image generation and inversion dominate the overall latency, taking more than one second. This demonstrates that the proposed watermarking procedure adds negligible overhead to the generation pipeline while maintaining strong robustness.
>
> ---
>
> We appreciate the reviewer’s insightful comments and believe the revisions have further improved the clarity and strength of the paper. Should you have any further questions, we are more than happy to discuss.

---

> ### Comment · Reviewer_wv9s · 2025-12-16
>
> I thank the authors for their responses and I don't have further questions.

---

> > ### Author Response · Authors · 2025-12-16
> >
> > We sincerely thank the reviewer for their careful review and constructive feedback. We are glad that our responses have addressed the concerns.

---

### Decision · Action_Editor_MvBe · 2026-01-07

**Recommendation:** Accept with minor revision

**Additional Comments:**

Required changes

- Confusing presentation of the watermark generation. In Sec. 4.1, only the conditional density of $z_T^e$ is defined, not how the actual $z_T^e$ is generated. Given that different random variables can have the same distribution and hence density function, this doesn't specify how the code is actually generated from an algorithm point of view. Can you be more explicit about this? The current AE suspects that once this is spelled out, the proof of Lemma 1 (and also the multi-dimensional extension) actually becomes trivial

- Better presentation for analysis of patch aggression robustness. Similar to the above point, in Sec 4.3, the essence of Proposition 1 is a Gaussian approximation to a binomial distribution, which is quite standard in elementary probability and hence the proof can be omitted. The current AE suggests clarifying the simple intuition behind this Proposition to help the reader understand what happens intuitively, rather than overwhelming them with unnecessary mathematics.

**Audience:**

Yes

**Audience Explanation:**

Watermarking is a promising way for copyright protection and source tracing for generative models, and is hence of interest to the community

**Claims And Evidence:**

Yes

**Claims Explanation:**

This paper introduces DiffKGW, a training-free watermarking method that adapts KGW techniques for LLMs to the continuous latent space of diffusion models. Its strengths include a mathematically sound approach that preserves Gaussian distributions, high visual fidelity, and strong robustness against geometric and noise-based attacks. Reviewers also like the paper's new taxonomy of design dimensions for watermarks. Initial concerns centered around weaknesses stemming from limited novelty, given its (trivial?) generalization of existing watermark mechanisms for LLMs and the marginal practical gains given its complex hybrid strategy. Most of these concerns were resolved through detailed rebuttal and expanded failure analyses

---

> ### Author Response · Authors · 2026-02-10
>
> Dear AC,
>
> We sincerely thank the AC for the valuable comments and careful coordination throughout the review process. We truly appreciate the time and effort invested in helping improve the paper.
>
> In Sec. 4.1, we have added an explicit description of how $z_T^e$ is generated, clarifying the watermark generation process beyond the conditional density. In Sec. 4.3 Proposition 1, we omit the detailed proof and instead present the key insights of the analysis to convey the intuition behind patch aggregation robustness, avoiding unnecessary complexity.
>
> We have revised the manuscript to address the raised concerns and improve clarity and presentation, and have submitted the updated camera-ready version. Thank you again for the constructive feedback and support.
>
> Best,
>
> Authors